# ADVERSARIAL ATTACK ON TENSOR RING DECOMPOSITION

## ABSTRACT

Tensor ring (TR) decomposition, a powerful tool for handling high-dimensional data, has been widely applied in various fields such as computer vision and recommender systems. However, the vulnerability of TR decomposition to adversarial perturbations has not been systematically studied, and it remains unclear how adversarial perturbations affect its low-rank approximation performance. To tackle this problem, we introduce a novel adversarial attack approach on tensor ring decomposition (AdaTR), formulated as an asymmetric max–min objective. Specifically, we aim to find the optimal perturbation that maximizes the reconstruction error of the low-TR-rank approximation. Furthermore, to alleviate the memory and computational overhead caused by iterative dependency during attacks, we propose a novel faster approximate gradient attack model (FAG-AdaTR) that avoids step-by-step perturbation tensor tracking while maintaining high attack effectiveness. Subsequently, we develop a gradient descent algorithm with numerical convergence guarantees. Numerical experiments on tensor decomposition, completion, and recommender systems using color images and videos validate the attack effectiveness of the proposed methods.

## 1 INTRODUCTION

Tensor decompositions aim to decompose the higher-order tensor to a set of low dimensional factors, which have attracted significant attention in various fields, including machine learning (Kolda & Bader, 2009), quantum physics (Sidiropoulos et al., 2017), signal processing (Schütt et al., 2020), brain science (Kang et al., 2013), and chemometrics (Acar et al., 2011). Different from matrix decomposition, there is no unique definition for the corresponding tensor decomposition. The CANDECOMP/PARAFAC (CP) decomposition (Hitchcock, 1927) can be regarded as a special case of the Tucker (Tucker, 1966) decomposition, where the core factor has nonzero entries only on the super-diagonal. However, Tucker decomposition suffers from restrictive bounds on its Tucker ranks, limiting its ability to capture rich structural information in high-order tensors. To address this issue, tensor train (TT) (Oseledets, 2011) and tensor ring (TR) (Zhao et al., 2016) decompositions have been proposed and have shown strong performance on tensor decomposition tasks. Specifically, TT decomposition represents an $N$th-order tensor using $(N-2)$ third-order tensors and two matrices, while TR decomposition factorizes it into $N$ third-order tensors. TT can further be viewed as a special case of TR, where the border tensor ranks are constrained to one.

Recently, Goodfellow et al. (2014) demonstrated that machine learning methods are vulnerable to adversarial attacks and has been widely verified in various fields (Ebrahimi et al., 2017; Zou et al., 2023; Wang et al., 2025). Motivated by this, adversarial training for the nonnegative matrix factorization (ANMF) model has been investigated to improve the robustness and predictive performance of NMF (Luo et al., 2020). However, their formulation does not make it easy to choose the instance-specific target. Therefore, Cai et al. (2021) proposed the novel adversarially-trained NMF (ATNMF) to tackle this problem, which can be written as follows:

$$\min_{\mathbf{W}, \mathbf{H} \geq 0} \|\mathbf{X} + \mathbf{E} - \mathbf{W}\mathbf{H}\|_{\mathrm{F}}^2$$

$$\text{s.t. } \mathbf{E} = \arg\max_{\mathbf{E}} \|\mathbf{X} + \mathbf{E} - \mathbf{W}\mathbf{H}\|_{\mathrm{F}}^2, \mathbf{X} + \mathbf{E} \geq 0, \|\mathbf{E}\|_{\mathrm{F}}^2 < \epsilon, \quad (1)$$

where the $\mathbf{X} \in \mathbb{R}^{I \times J}$ denotes the original data matrix, and $\mathbf{W} \in \mathbb{R}^{I \times R}, \mathbf{H} \in \mathbb{R}^{R \times J}$ denote the nonnegative factors, and $\mathbf{E} \in \mathbb{R}^{I \times J}$ denotes the perturbation matrix. The $\epsilon$ denotes the energy budget

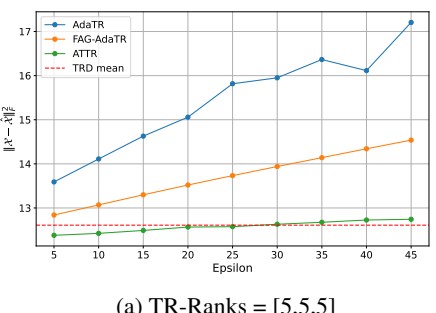 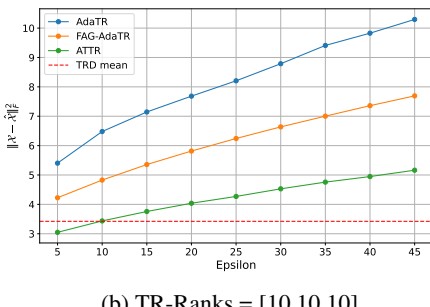

(a) TR-Ranks = [5,5,5]     (b) TR-Ranks = [10,10,10]

Figure 1: Average reconstruction error $\|\mathcal{X} - \hat{\mathcal{X}}\|_\text{F}^2$ under different perturbation budget $\epsilon$ for (a) Rank = 5 and (b) Rank = 10. Results are averaged over eight images. The proposed methods (AdaTR and FAG-AdaTR) are compared with ATTR and the TR decomposition baseline.

of the perturbation matrix $\mathbf{E}$, and $\|\cdot\|_\text{F}$ denotes the Frobenius norm. This approach is particularly beneficial across various real-world applications, include matrix completion, link prediction, recommender systems (Seyedi et al., 2023; Mahmoodi et al., 2023; 2024; Zhang et al., 2024). When a limited budget $\epsilon$ is given, the approach Eq. (1) can usually improve the predictive performance rather than attacking matrix factorization (Zhang et al., 2024).

Compared to employing adversarial training to improve the robustness of matrix factorization, verifying whether matrix factorization itself is inherently vulnerable is more challenging. Related studies include maliciously injecting outlier samples into the matrix (i.e., $\mathbf{X}_\text{adv} = [\mathbf{X}, \mathbf{z}]$) to cause the attacked subspace to deviate from the original data subspace (Pimentel-Alarcón et al., 2017; Li et al., 2021), adding adversarial perturbations that lead to subspace deviation (Li et al., 2020), or breaking the uniqueness of NMF (Vu et al., 2024). These studies have mainly focused on subspace deviation or on compromising the uniqueness of the factorization. However, the vulnerability of the most general matrix or tensor decomposition remains an open question. This motivates us to ask the following question:

- **RQ1:** Are matrix/tensor factorization vulnerable to adversarial attacks?

- **RQ2:** How can we design the adversarial attack approach for matrix/tensor decompositions?

To address these questions, we first extend the concept of ATNMF to TR decomposition, yielding an ATTR baseline approach. Here, vulnerability refers to which a perturbation added to the input tensor, within a given budget, can increase the resulting reconstruction error. As illustrated in Fig.1, ATTR exhibits behavior consistent with ATNMF: under small perturbations, ATTR slightly improves predictive performance. However, when evaluated under the same perturbation budget, our proposed methods (AdaTR and FAG-AdaTR) lead to substantially larger low-rank approximation errors. This provides clear evidence that TR decomposition is truly vulnerable to adversarial perturbations, thereby answering **RQ1**. All ALS-based tensor decompositions—whether for reconstruction, completion, or recommendation—are entirely driven by the observed input tensor, even small perturbations injected at the input propagate through all update steps and get amplified by the low-rank structure, ultimately causing large reconstruction errors (and consequently large prediction errors in recommendation).

Having established the vulnerability of TR decomposition, we next turn to **RQ2**. To this end, we propose a novel asymmetric adversarial attack approach for TR decomposition, termed AdaTR. In particular, we define the low-rank approximation error as the attack objective and model the perturbation as a learnable adversarial perturbation tensor. Different from the traditional adversarial training approach as in Eq. (1), the proposed AdaTR adopts an asymmetric max-min objective. This design enables the attacker to directly maximize the low-rank approximation error in TR decomposition. Furthermore, to mitigate the high computational overhead of long iterative dependencies, we introduce FAG-AdaTR, a faster attack algorithm with an approximate gradient. The key contributions of this work are summarized as follows:

- We elaborately design an asymmetric adversarial attack approach on TR decomposition (AdaTR). This approach provides the first evidence that tensor decomposition models are susceptible to adversarial attacks.
- AdaTR requires backtracking TR iterative updates, demanding substantial peak memory. To alleviate this problem, we propose a faster algorithm with approximate gradient on TR decomposition (FAG-AdaTR).
- Extensive experiments show that our proposed attacks substantially degrade performance in tensor decomposition, completion, and recommendation tasks, and are capable of causing significant errors even with tiny perturbations.

## 2 NOTATIONS

In this paper, the scalars are denoted by standard lowercase or uppercase letters (e.g., $x$, $X$), vectors by bold lowercase letters (e.g., $\mathbf{x}$), and matrices by bold uppercase letters (e.g., $\mathbf{X}$). Higher-order tensors with order $N \geq 3$ are denoted by calligraphic letters, e.g., $\mathcal{X} \in \mathbb{R}^{I_1 \times I_2 \times \cdots \times I_N}$. The $(i_1, i_2, \cdot, i_N)$-th element of $\mathcal{X}$ is denoted by $\mathcal{X}(i_1, i_2, \ldots, i_N)$ or equivalently $x_{i_1, i_2, \ldots, i_N}$. The Frobenius norm of a tensor is defined as $\|\mathcal{X}\|_{\mathrm{F}} = \sqrt{\sum_{i_1, i_2, \cdots, i_N} \mathcal{X}(i_1, i_2, \cdots, i_N)}$. The set notation is denoted by $[\mathcal{G}] := \{\mathcal{G}_1, \mathcal{G}_2, \cdots, \mathcal{G}_N\}$.

## 3 PRELIMINARIES

**Definition 1** (Tensor Composition)**.** *We call the process of generating the $N$-th order tensor $\mathcal{X}$ from the factors $\{\mathcal{G}_1, \mathcal{G}_2, \cdots, \mathcal{G}_N\}$ in special tensor network contraction as the tensor composition, which can be written as $\mathcal{X} = TN([\mathcal{G}])$. Furthermore, we can also write the tensor composition except the factor $\mathcal{G}_k$ as $TN(\{\mathcal{G}_1, \cdots, \mathcal{G}_{k-1}, \mathcal{G}_{k+1}, \cdots, \mathcal{G}_N\})$ or $TN([\mathcal{G}], /\mathcal{G}_k)$.*

**Definition 2** (Tensor Decomposition)**.** *We call the process of learning the $N$ factors $\mathcal{G}$ of the $N$-th order tensor $\mathcal{X}$ in a specific tensor network method as tensor decomposition. The decomposition operator of the tensor $\mathcal{X}$ can be written as $\mathcal{X} \approx TN([\mathcal{G}])$.*

**Definition 3** (Tensor Ring Decomposition (Zhao et al., 2016))**.** *The TR decomposition representation is given as follows,*

$$\mathcal{X}(i_1, i_2, \cdots, i_N) = \sum_{r_1, \cdots, r_N}^{R_1, \cdots, R_N} \mathcal{G}_1(r_1, i_1, r_2)\mathcal{G}_2(r_2, i_2, r_3) \cdots \mathcal{G}_N(r_N, i_N, r_1), \tag{2}$$

*where $\mathcal{X} \in \mathbb{R}^{I_1 \times I_2 \times \cdots \times I_N}$ denotes an $N$th-order tensor, and $\mathcal{G}_n \in \mathbb{R}^{R_n \times I_n \times R_{n+1}}$ are third-order factors. Symbolically, we employ $\mathcal{X} = TR([\mathcal{G}]) = TR(\mathcal{G}_1, \mathcal{G}_2, \cdots, \mathcal{G}_N)$ to denote TR decomposition.*

**Definition 4** (Tensor Mode-$k$ Unfolding (Zhao et al., 2016))**.** *Given an $N$th-order tensor $\mathcal{X} \in \mathbb{R}^{I_1 \times I_2 \times \cdots I_N}$, the tensor mode-$k$ unfolding of $\mathcal{X}$ is given as follows:*

$$\mathbf{X}_{[k]}(i_k, \overline{i_{k+1}i_{k+2}\cdots i_N i_1 i_2 \cdots i_{k-1}}) = \mathcal{X}(i_1, i_2, \cdots, i_N), \tag{3}$$

*and the classical mode-$k$ unfolding of $\mathcal{X}$ is given as follows:*

$$\mathbf{X}_{(k)}(i_k, \overline{i_1 i_2 \cdots i_{k-1} i_{k+1} i_{k+2} \cdots i_N}) = \mathcal{X}(i_1, i_2, \cdots, i_N), \tag{4}$$

*where $\mathbf{X}_{[k]}$ and $\mathbf{X}_{(k)}$ are the size of $I_k \times \prod_{j \neq k} I_j$ matrices.*

## 4 ADVERSARIAL ATTACK ON TENSOR RING DECOMPOSITION

### 4.1 WHY WE NEED AN ASYMMETRIC ADVERSARIAL ATTACK FRAMEWORK ON TENSOR DECOMPOSITION

The ATNMF algorithm can be naturally extended to tensor decomposition. As an example, we consider the tensor ring (TR) decomposition, which allows us to use this adversarial training approach

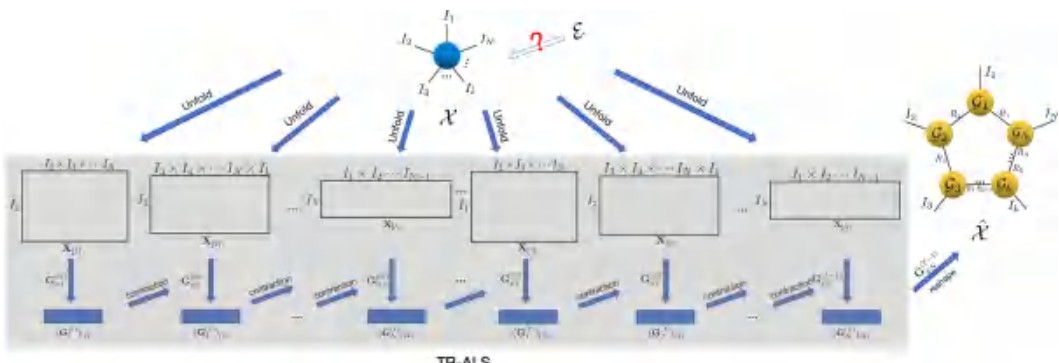

Figure 2: Illustration of TR-ALS algorithm. If the adversarial perturbation tensor $\mathcal{E}$ is added to the input $\mathcal{X}$, it propagates through each unfolding and update step, eventually leading to a perturbed reconstruction $\hat{\mathcal{X}}$.

to it (ATTR). Specifically, we can describe ATTR as follows:

$$\max_{\|\mathcal{E}\|_F^2 \leq \epsilon} \min_{[\mathcal{G}]} \frac{1}{2}\|\mathcal{X} + \mathcal{E} - \mathrm{TR}([\mathcal{G}])\|_F^2. \tag{5}$$

However, this symmetric min–max objective suffers from a fundamental limitation. In practice, the update of $\mathcal{E}$ does not explicitly maximize the reconstruction error of TR decomposition; instead, it degenerates into maximizing the difference between successive perturbations, i.e., $\|\mathcal{E}^{(t)} - \mathcal{E}^{(t-1)}\|_F^2$. As a result, ATTR cannot guarantee that the perturbation $\mathcal{E}$ effectively degrades the decomposition performance. In fact, under tiny perturbations, ATTR may even *improve* the predictive performance of TR decomposition. This somewhat counterintuitive phenomenon can be explained theoretically as follows.

**Theorem 1.** *Let $\delta$ be the reconstruction error of standard TR-ALS algorithm, and $\mathcal{R}_2$ be the residual term of ATTR algorithm. If the perturbation budget $\epsilon$ satisfies:*

$$\sqrt{\epsilon} < \sqrt{\delta} - \|\mathcal{R}_2\|_F, \tag{6}$$

*then ATTR achieves a smaller reconstruction error than standard TR-ALS.*

The proof is deferred to Appendix B.

**Remark 1.** *This result shows that when the perturbation strength $\epsilon$ is sufficiently small relative to the gap between the TR-ALS error bound $\delta$ and the ATTR residual $\|\mathcal{R}_2\|_F^2$, adversarial training can improve predictive performance instead of degrading it.*

This paradoxical behavior demonstrates the inherent limitation of ATTR: it does not ensure that perturbations effectively attack the decomposition. This limitation motivates the need for an asymmetric adversarial attack approach to maximize the low-rank approximation error on the tensor decomposition.

When we focus on minimizing the factors $[\mathcal{G}]$ of Eq. (5), this model can be formulated as follows:

$$\min_{[\mathcal{G}]} \frac{1}{2}\|\mathcal{X} + \mathcal{E} - \mathrm{TR}([\mathcal{G}])\|_F^2, \tag{7}$$

where its closed-form solution can be obtained by using the ALS algorithm:

$$(\mathbf{G}_n^{(t)})_{(2)} = (\mathbf{X}_{[n]} + \mathbf{E}_{[n]})\mathbf{G}_{\neq n}^{(t-1)\dagger}, \tag{8}$$

where the $\mathbf{G}_{\neq n}$ denotes the unfold tensor composed by $[\mathcal{G}]$ without factor $\mathcal{G}_n$, and $(\mathbf{G}_n)_{(2)}$ denotes the mode-2 unfolding of $\mathcal{G}_n$. It is clear that the update of $\mathcal{G}_n$ inevitably involves the perturbation tensor $\mathcal{E}$. Consequently, the updated $\mathcal{G}_{\neq n}^{(t)}$, $n = 1, 2, \ldots, N$ is contaminated by $\mathcal{E}$, and therefore $\mathcal{G}$ can be regarded as a function of adversarial perturbation $\mathcal{E}$:

$$\mathcal{G}_n(\mathcal{E}) := f_n(\mathcal{E}, \mathcal{G}_{\neq k}; \mathcal{X}), \tag{9}$$

where $f_n(\cdot)$ denotes an intrinsic function that mapping the $\mathcal{E}$ and $\mathcal{G}_{\neq n}$ to $\mathcal{G}_n$. Fig. 2 illustrates the gradient flow between the core factors and adversarial perturbation $\mathcal{E}$ during the update process.

## 4.2 ADATR ATTACK ALGORITHM

Intuitively, the attacker injects a small, bounded perturbation $\mathcal{E}$ into the observed tensor $\mathcal{X}$. In the traditional symmetric min-max approach as in Eq. (1), the defender (low-rank approximation using TR-ALS) usually estimates the core factors by minimizing the approximation error on the perturbed tensor $\mathcal{X} + \mathcal{E}$. And then the attacker will maximize $\|\mathcal{X} + \mathcal{E} - \mathrm{TR}([\mathcal{G}^{(t-1)}])\|_{\mathrm{F}}^2$ with fixed $[\mathcal{G}]$. However, this deviation does not match the attack goal for tensor decomposition, i.e., to maximize low-rank approximation error.

To this end, we propose an asymmetric max-min objective to maximize the low-rank approximation error with respect to $\mathcal{E}$ via the intrinsic function $f_n$ in Eq. (9), while concurrently minimizing the approximation error with respect to the core factors $[\mathcal{G}]$ using a standard low-rank approximation procedure. Formally, the attacker is modeled by the following bilevel optimization:

$$\max_{\mathcal{E}} \frac{1}{2}\|\mathcal{X} - \mathrm{TR}([\mathcal{G}^{(T)}(\mathcal{E})])\|_{\mathrm{F}}^2,$$

$$\text{s.t. } [\mathcal{G}^{(T)}(\mathcal{E})] = \arg\min_{[\mathcal{G}]} \frac{1}{2}\|\mathcal{X} + \mathcal{E} - \mathrm{TR}([\mathcal{G}])\|_{\mathrm{F}}^2, \quad \|\mathcal{E}\|_{\mathrm{F}}^2 < \epsilon, \tag{10}$$

where $[\mathcal{G}^{(T)}(\mathcal{E})]$ indicates that $\mathcal{G}_n^{(T)}$ is a function of $\mathcal{E}$, as defined in Eq. (9), and is obtained from the minimization problem at the $T$-th iteration of the TR decomposition.

To maximize Eq. (10) with respect to $\mathcal{E}$, we first let $g := \frac{1}{2}\|\mathcal{X} - \mathrm{TR}([\mathcal{G}^{(T)}(\mathcal{E})])\|_{\mathrm{F}}^2$, and combine with Eq. (9), we can calculate the gradient for $\mathcal{E}$ according to the chain rule:

$$\frac{\partial g}{\partial \mathcal{E}} = \frac{\partial g}{\partial f_N}\frac{\partial f_N}{\partial \mathcal{E}} + \frac{\partial g}{\partial f_{N-1}}\frac{\partial f_{N-1}}{\partial \mathcal{E}} + \cdots + \frac{\partial g}{\partial f_1}\frac{\partial f_1}{\partial \mathcal{E}}. \tag{11}$$

Since $\mathcal{E}$ is involved in the dependencies across different $\mathcal{G}$'s (as illustrated in Fig. 2), explicitly deriving the gradient expression with respect to $\mathcal{E}$ becomes extremely complex. Therefore, in practice, we compute the gradient of $\mathcal{E}$ using PyTorch's automatic differentiation engine and update using gradient ascent:

$$\mathcal{E}^{(t)} = \mathcal{E}^{(t-1)} + \eta\frac{\partial g}{\partial \mathcal{E}}, \tag{12}$$

until reaching the convergence conditions. We summarize the AdaTR algorithm in Algorithm 1.

Noting that adversarial training is not applicable in our setting, since tensor decompositions are non-parametric procedures that recompute factor tensors from scratch for each input and therefore do not retain trainable parameters for robustness learning. At the same time, although we instantiate the attack with TR decomposition, the bilevel formulation itself is general and can be directly applied to other ALS-based tensor models (e.g., CP, Tucker, TT).

## 4.3 FASTER APPROXIMATE GRADIENT ATTACK MODEL

The proposed AdaTR algorithm needs extensive backpropagation on the intrinsic function $f_n$, which imposes considerable computational overhead. To alleviate this problem, we introduce a faster approximate gradient strategy of the adversarial attack algorithm (FAG-AdaTR) in this subsection. Specifically, the method leverages only the gradient of factors $[\mathcal{G}^{(t=T)}]$ update, thereby reducing resource consumption while preserving the effectiveness of the optimization process.

According to the proposed Eq. (10), we can rewrite it as follows with matrix formulation in the $T$-th iteration:

$$\max_{\mathbf{E}_{[n]}} \frac{1}{2}\|\mathbf{X}_{[n]} - \mathbf{G}_{(2)}^{n(t=T)}(\mathcal{E})\mathbf{G}_{\neq n}^{(t=T)}(\mathcal{E})\|_{\mathrm{F}}^2. \tag{13}$$

To simplify the gradient calculation, we assume $\mathbf{G}_{\neq n}^{(t=T)}(\mathcal{E})$ is independent of $\mathcal{E}$. Therefore, Eq. (13) can be reformulated as

$$\max_{\mathbf{E}_{[n]}} \frac{1}{2}\|\mathbf{X}_{[n]} - \mathbf{G}_{(2)}^{n(t=T)}(\mathcal{E})\mathbf{G}_{\neq n}^{(t=T)}\|_{\mathrm{F}}^2. \tag{14}$$

However, the matrix $\mathbf{G}_{(2)}^{n(t=T)}(\mathcal{E})$ remains coupled with $\mathcal{E}$ across different iterations $t$, which makes its explicit formulation intractable. We further assume that both $\mathbf{G}_{(2)}^{n(t=T-1)}(\mathcal{E})$ and $\mathbf{G}_{\neq n}^{(t=T-1)}(\mathcal{E})$

are the variables independent of $\mathcal{E}$. Based on the above assumption, Eq. (14) can be rewritten as

$$\max_{\mathbf{E}_{[n]}} h_n(\mathbf{E}_{[n]}),  \tag{15}$$

where $h_n(\mathbf{E}_{[n]}) := \frac{1}{2}\|\mathbf{X}_{[n]} - (\mathbf{X}_{[n]} + \mathbf{E}_{[n]})\mathbf{G}_{\neq n}^{(t=T-1)\dagger}\mathbf{G}_{\neq n}^{(t=T)}\|_F^2$. Thus, the explicit gradient formulation with respect to the loss function Eq. (15) can be obtained directly:

$$\nabla_{\mathbf{E}_{[n]}} h_n(\mathbf{E}_{[n]}) = \left(\mathbf{X}_{[n]} - (\mathbf{X}_{[n]} + \mathbf{E}_{[n]})\mathbf{G}_{\neq n}^{(t=T-1)\dagger}\mathbf{G}_{\neq n}^{(t=T)}\right)\mathbf{G}_{\neq n}^{(t=T)\dagger}\mathbf{G}_{\neq n}^{(t=T-1)},  \tag{16}$$

which allows the gradient ascent update for $\mathcal{E}$:

$$\mathcal{E}^{(t)} = \mathcal{E}^{(t-1)} + \eta \sum_{n=1}^{N} \omega_n \text{Fold}_n(\nabla_{\mathbf{E}_{[n]}} h_n(\mathbf{E}_{[n]})),  \tag{17}$$

where $\omega_n = I_n/(\sum_j I_j)$ is denotes the mode-$n$ weight, and $\text{Fold}_n(\cdot) : \mathbb{R}^{I_n \times I_1 I_2 \cdots I_N} \to \mathbb{R}^{I_1 \times I_2 \times \cdots \times I_N}$ denotes the tensor folding operation.

In contrast to the proposed AdaTR algorithm, FAG-AdaTR reduces the dependency of the adversarial perturbation $\mathcal{E}$ on different iterations and different core tensors, thereby allowing the gradient to be computed more efficiently. The overall optimization procedure for FAG-AdaTR is summarized in Algorithm 2.

### 4.4 CONVERGENCE ANALYSIS

In this subsection, we first establish convergence Theorem 2 of AdaTR. Lemma 1 then clarifies AdaTR cannot exhibit the collapse behavior observed in ATTR based on Theorem 2. Finally, Theorems 3-4 extend the convergence guarantees to the FAG-AdaTR.

**Theorem 2** (Convergence of AdaTR). *Suppose that assumptions: (i) the map $\mathcal{E} \mapsto [\mathcal{G}^{(T)}(\mathcal{E})]$ is differentiable on $\mathcal{B} = \{\mathcal{E} : \|\mathcal{E}\|_F^2 \leq \epsilon\}$ with bounded Jacobian; (ii) the TR reconstruction $\text{TR}(\cdot)$ is smooth on bounded sets. The proposed AdaTR stepsizes satisfy $0 < \underline{\eta} \leq \eta_t \leq \bar{\eta} \leq 1/L$ for all $t$. Then the sequence $\{\mathcal{E}^{(t)}\}$ generated by Alg. (1) has the following conclusions:*

*1. The objective values are monotonically nondecreasing.*

*2. The sequence $\{\mathcal{E}^{(t)}\}$ is the Cauchy sequence.*

*3. Any limit point of sequence $\{\mathcal{E}^{(t)}\}$ statisfies the KKT conditions of problem (10).*

Detailed lemmas and proofs can be found in Appendix C.

**Lemma 1** (Monotonicity prevents collapse). *Assume (i) Theorem 1 holds so that the perturbation $\widetilde{\mathcal{E}}$ produced by ATTR satisfies $g(\widetilde{\mathcal{E}}) < g(\mathbf{0})$, and (ii) AdaTR is initialized at $\mathcal{E}^{(0)}$ with $g(\mathcal{E}^{(0)}) > g(\mathbf{0})$. Then, by the monotonic ascent property of AdaTR (Theorem 2), we have*

$$g(\mathcal{E}^{(t)}) \geq g(\mathcal{E}^{(0)}) > g(\mathbf{0}) > g(\widetilde{\mathcal{E}}), \qquad \forall t \geq 0,  \tag{18}$$

*where the $\widetilde{\mathcal{E}}$ is the perturbation generated by ATTR, $g(\mathbf{0})$ is the reconstruction error of clean tensor. Thus, in the regime where ATTR reduces the reconstruction error of $\mathcal{X}$, AdaTR always increases it and therefore cannot exhibit the same collapse behavior reported in ATTR.*

The proof is provided in Appendix C.3.

**Theorem 3** (Convergence of FAG-AdaTR). *Suppose that the TR factors $\{\mathbf{G}_{\neq n}^{(T-1)}, \mathbf{G}_{\neq n}^{(T)}\}_{n=1}^{N}$ remain bounded on the perturbation ball $\mathcal{B} = \{\mathcal{E} : \|\mathcal{E}\|_F^2 \leq \epsilon\}$, and that the step sizes satisfy $0 < \underline{\eta} \leq \eta_t \leq \bar{\eta} \leq 1/L$ for all $t$, where $L > 0$ is a Lipschitz constant of $\nabla\tilde{g}$ on $\mathcal{B}$. Then the sequence $\{\mathcal{E}^{(t)}\}$ generated by Alg. 2 satisfies:*

*1. The objective values are monotonically nondecreasing.*
*2. The sequence $\{\mathcal{E}^{(t)}\}$ is Cauchy, and hence convergent in $\mathcal{B}$.*
*3. Any limit point $\mathcal{E}^\star$ of $\{\mathcal{E}^{(t)}\}$ satisfies the KKT stationarity conditions for the projected maximization problem (15)*

The proof follows standard arguments for projected gradient methods on smooth objectives and is deferred to Appendix D.

**Theorem 4** (Approximate stationarity of FAG-AdaTR). *Let $\mathcal{E}^\star$ be any limit point of FAG-AdaTR. Assume that $\|\nabla g(\mathcal{E}) - \nabla \tilde{g}(\mathcal{E})\|_F \leq \varepsilon_g$ holds on $\mathcal{B} = \{\mathcal{E} : \|\mathcal{E}\|_F^2 \leq \epsilon\}$. Then*

$$\langle \nabla g(\mathcal{E}^\star), \mathcal{Y} - \mathcal{E}^\star \rangle \geq -\sqrt{\epsilon}\, \varepsilon_g, \qquad \forall \mathcal{Y} \in B. \tag{19}$$

*Hence, $\mathcal{E}^\star$ is an $O(\sqrt{\epsilon}\,\varepsilon_g)$-approximate stationary point of the true objective $g(\mathcal{E})$.*

The proof is provided in Appendix E.

## 5 EXPERIMENTAL RESULTS

In this section, we compare the proposed methods with the ATTR method under the defense of different TR decomposition methods. Specifically, the defense baselines include TR-ALS (Zhao et al., 2016), TRPCA-TNN (Lu et al., 2019), TRNNM (Yu et al., 2019), HQTRC (He & Atia, 2022), and LRTC-TV (Li et al., 2017). We evaluate the proposed methods on three types of tensor data (color images, videos, and recommender system datasets) and test them across three representative tasks: tensor decomposition, tensor completion, and recommendation. Implementation details can be found in the Appendix.

### 5.1 COLOR IMAGES DECOMPOSITION ATTACK

In this subsection, we evaluate adversarial attacks on color image decomposition using tensor-based defense methods. The eight widely-used color images are chosen from the DIV2K dataset[1] for testing data, and each color image is of the size $672 \times 1020 \times 3$ and normalized into [0, 1].

Table 1: RSE matrix: mean $\pm$ variance across runs. Lower is better; **bold** marks the worst (highest RSE) per defense (column).

| Attack \ Defense | TR-ALS | TRPCA-TNN | TRNNM | HQTRC-Cor | HQTRC-Cau | HQTRC-Hub | LRTC-TV |
|---|---|---|---|---|---|---|---|
| Clean | $0.202 \pm 0.010$ | $0.111 \pm 0.001$ | $0.152 \pm 0.001$ | $0.163 \pm 0.006$ | $0.154 \pm 0.005$ | $0.126 \pm 0.002$ | $0.147 \pm 0.003$ |
| Gauss Noise | $0.230 \pm 0.008$ | $0.376 \pm 0.005$ | $0.546 \pm 0.011$ | $0.346 \pm 0.005$ | $0.330 \pm 0.004$ | $0.377 \pm 0.005$ | $0.271 \pm 0.003$ |
| ATTR-gen | $0.289 \pm 0.010$ | $0.554 \pm 0.010$ | $0.556 \pm 0.011$ | $0.351 \pm 0.005$ | $0.346 \pm 0.005$ | $0.360 \pm 0.005$ | $0.300 \pm 0.004$ |
| AdaTR-gen | $\mathbf{0.794 \pm 0.025}$ | $0.681 \pm 0.020$ | $0.680 \pm 0.016$ | $0.453 \pm 0.011$ | $0.444 \pm 0.011$ | $0.465 \pm 0.010$ | $0.340 \pm 0.005$ |
| FAG-AdaTR-gen | $0.744 \pm 0.020$ | $\mathbf{0.703 \pm 0.018}$ | $\mathbf{0.686 \pm 0.017}$ | $\mathbf{0.564 \pm 0.018}$ | $\mathbf{0.556 \pm 0.018}$ | $\mathbf{0.578 \pm 0.017}$ | $\mathbf{0.560 \pm 0.017}$ |

Tab. 1 shows all methods' average RSE values over eight color images. The best results are highlighted in bold. It can be seen that the proposed methods achieve superior results to the ATTR in all cases. Especially in the color Fig. 8 of Appendix, we can see that the reconstructed image of the proposed AdaTR attack on TR-ALS makes the person indistinguishable to the human eye. Although our attack was performed only against the TR-ALS algorithm, the adversarial images generated by attacking TR-ALS transfer to all tested TR-based defense methods and consistently produce the strongest attack results.

### 5.2 VIDEOS DECOMPOSITION ATTACK

In this subsection, we evaluate the effectiveness of the proposed method on color video data[2] for the tensor decomposition task. For fair comparison, we randomly select the seven color videos, and each video in the dataset consists of at least 150 frames. Moreover, Zhou et. al (Zhou et al., 2017) find that the color video *news* of 30 frames has much more redundant information in their experiment. Thus, we select the ten consistent frames for each color video. Each video segment is thus represented as a fourth-order tensor of size $144 \times 176 \times 3 \times 10$ (spatial height $\times$ spatial width $\times$ color channel $\times$ frame).

Fig. 3 presents the evaluation results of the average PSNR, SSIM, RSE of all methods. The numerical comparison in Fig. 3 clearly demonstrates the superiority of the proposed methods. The

---

[1] https://data.vision.ee.ethz.ch/cvl/DIV2K/
[2] http://trace.eas.asu.edu/yuv/

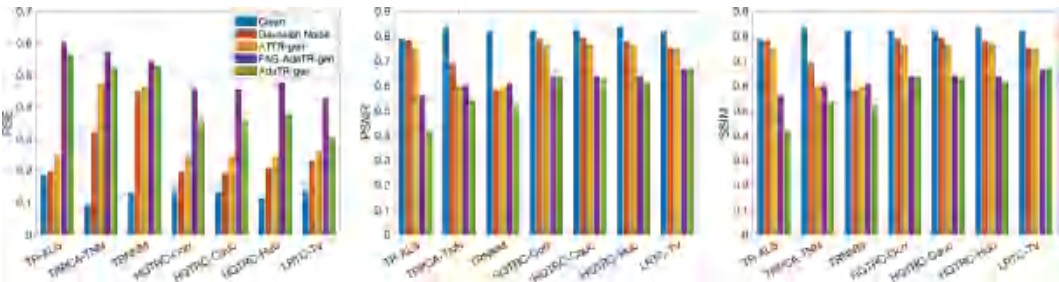

Figure 3: Average RSE, PSNR, and SSIM results over seven videos on tensor decomposition tasks.

proposed methods achieve approximately 2 times gain in average RSE compared to TR-ALS, which demonstrates its competitive advantage in terms of the attack on tensor decomposition. To facilitate a more comprehensive visual comparison, Fig. 20 presents the reconstruction results for the 5th frame of the color video *news*. It can be clearly observed that the proposed method is able to attack local details and destroy the global structure.

### 5.3 TENSOR COMPLETION ATTACK

In this subsection, we conduct experiments on color images to assess the effectiveness of the proposed method for the tensor completion task. The same testing data used as subsection 5.1, and each color image is normalized into [0, 1]. Fig. 4 presents the TC results for the randomly chosen color image with a sampling rate of 0.2. It can be clearly observed that the reconstructed image of the proposed AdaTR attack on TR-ALS makes the person indistinguishable to the human eye.

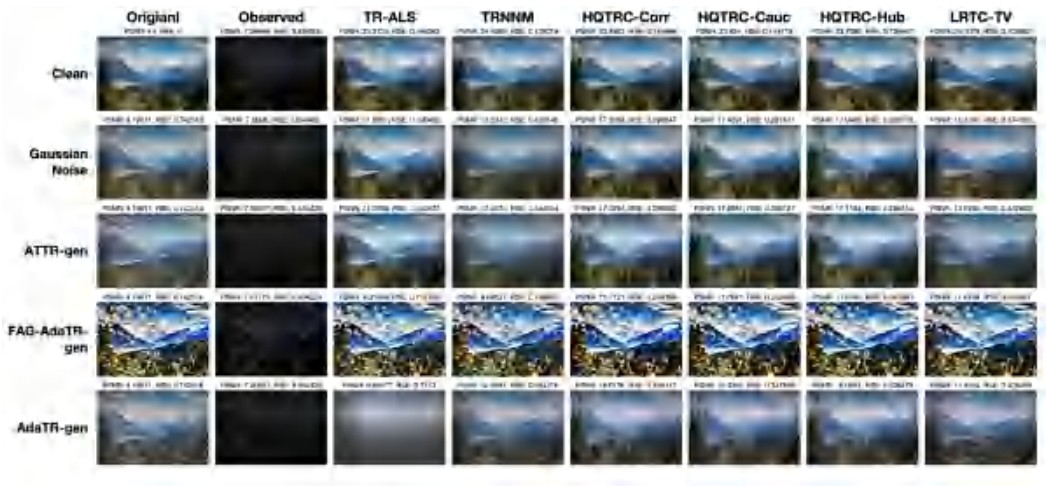

Figure 4: Visual example on tensor completion tasks under different attacks and defenses. Results are shown for one sample image (other seven cases are provided in the Appendix H).

### 5.4 RECOMMENDER SYSTEMS DECOMPOSITION ATTACK

In this subsection, we conduct experiments on a recommender systems dataset to assess the effectiveness of the proposed method for the recommendation task. To validate the recommendation performance, we extend the proposed AdaTR attack algorithm to the NMF model, termed AdaNMF. We use two datasets, including a synthetic dataset and the widely-used MovieLens-100K dataset[3]. The synthetic dataset is generated by randomly sampling 500 users and 450 items, with ratings ranging from 1 to 5, and we sample only 12% of the entries as observations. The MovieLens-100K dataset consists of 943 users and 1682 items, with about 6% of the user–item pairs observed. All training, testing, and perturbation operations are performed strictly on these observed entries. We preprocess

---

[3]https://grouplens.org/datasets/movielens/100k/

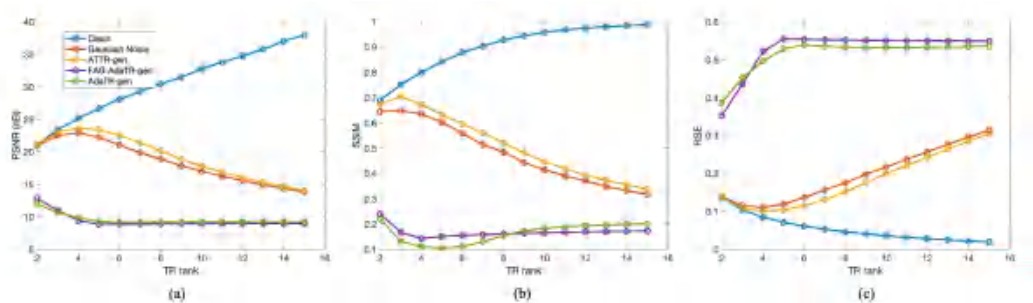

Figure 5: Adversarial attack in TR-ALS decomposition with different ranks defend. (a) PSNR, (b) SSIM, and (c) RSE results over a color image on tensor decomposition tasks.

the datasets by normalizing the ratings to the range [1, 5] and splitting them into training (80%) and testing (20%) sets. Regarding the NMF ranks in the proposed method, we set $R = 20$. Moreover, the values of $\epsilon$ and $\eta$ are set to 10 and 0.05, respectively. And **inner_num** and **outer_num** are fixed to 200 and 100 in all experiments.

Table 2: NMF recommender performance under different perturbations on Synthetic and MovieLens-100K datasets. Columns indicate desired direction: RMSE ($\downarrow$ better), Precision@10 ($\uparrow$ better), Recall@10 ($\uparrow$ better). **Bold** highlights effective attacks (AdaNMF) where RMSE increases and Precision/Recall decreases vs. Clean.

| Condition | Synthetic | | | MovieLens-100K | | |
|---|---|---|---|---|---|---|
| | RMSE $\uparrow$ | P@10 $\downarrow$ | R@10 $\downarrow$ | RMSE $\uparrow$ | P@10 $\downarrow$ | R@10 $\downarrow$ |
| Clean | 3.718 | 0.0132 | 0.0423 | 3.967 | 0.0200 | 0.0992 |
| Gaussian noise | 3.726 | 0.0124 | 0.0333 | 3.983 | 0.0153 | 0.0766 |
| ATNMF | **4.026** | 0.0142 | 0.0502 | **4.089** | 0.0123 | 0.0728 |
| **AdaNMF** | 4.024 | **0.0089** | **0.0266** | 4.078 | **0.0076** | **0.0461** |

Tab. 2 presents the evaluation results of the RMSE, Precision@10, and Recall@10 of all methods. The best results are highlighted in bold. It can be seen that the proposed AdaNMF method achieves superior results to the ATNMF in most cases.

## 5.5 TR-RANKS ROBUSTNESS OF ATTACKS

In this subsection, we conduct experiments to evaluate the robustness of the proposed methods against different TR-ranks. We randomly select one of the color images from the DIV2K dataset as the testing data. Fig. 5 shows the RSE, PSNR, and SSIM values of the reconstructed image under different TR-ranks. It can be seen that the proposed methods achieve superior robustness results compared to the ATTR in all cases. Noting that we only attack the TR-ALS algorithm with the same TR-ranks $R_1 = R_2 = R_3 = 5$ in all the attack methods.

## 5.6 JPEG, PNG IMAGE DEFENDING

In this subsection, we evaluate the effectiveness of the proposed methods under two common image compression formats: PNG and JPEG, since encoding/decoding may partially remove small-magnitude adversarial perturbations. We randomly select one of the color images from the DIV2K dataset as the testing data. Tab. 3 shows the RSE, PSNR, and SSIM values of the reconstructed image under different image storage formats. It can be seen that the JPEG and PNG image storage formats have little effect on the performance of the proposed methods.

## 5.7 HYPERPARAMETER EXPERIMENT

In this subsection, we conduct experiments to evaluate the impact of the hyperparameter $\epsilon$ on the performance of the proposed methods. We randomly select eight of the color images from the

Table 3: Comparison between **PNG** and **JPEG** image compression formats.

| Method | PNG | | | JPEG | | |
|---|---|---|---|---|---|---|
| | RSE ↓ | PSNR ↑ | SSIM ↑ | RSE ↓ | PSNR ↑ | SSIM ↑ |
| Clean | 0.087 | 25.749 | 0.839 | 0.088 | 25.648 | 0.835 |
| Gaussian Noise | 0.154 | 20.818 | 0.682 | 0.165 | 20.234 | 0.669 |
| ATTR-gen | 0.174 | 19.744 | 0.657 | 0.186 | 19.174 | 0.642 |
| FAG-AdaTR-gen | 0.348 | 13.725 | 0.326 | 0.334 | 14.083 | 0.337 |
| AdaTR-gen | **0.521** | **10.229** | **0.156** | **0.495** | **10.667** | **0.175** |

DIV2K dataset as the testing data. As shown in Fig. 1, our methods are effective even at small perturbation budgets, while ATTR can actually improve reconstruction performance when $\epsilon$ is very small.

### 5.8 CONVERGENCE ANALYSIS

In this subsection, we experimentally analyze the numerical convergence behaviour to verify the convergence of the proposed methods. Fig. 6 illustrates the average reconstructed error value curves of the eight color images with $\epsilon = 66$. We can observe that the loss function value converges to a specific value in the end, which implies that the proposed method is convergent numerically. The experimental results support the efficacy of the proposed methods in achieving convergence and validate their usefulness in practical scenarios.

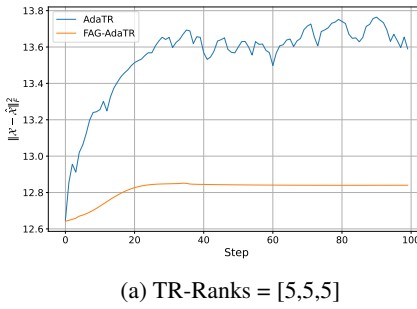
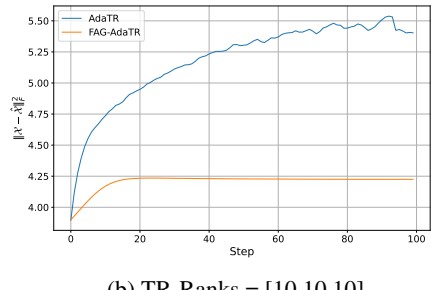

(a) TR-Ranks = [5,5,5]        (b) TR-Ranks = [10,10,10]

Figure 6: Convergence of the proposed AdaTR and FAG-AdaTR methods in terms of reconstruction error, averaged over 8 images decomposition with perturbation budget $\epsilon = 66$. Results are shown for (a) TR-Ranks = [5,5,5] and (b) TR-Ranks = [10,10,10].

## 6 CONCLUSION

This paper proposes a novel asymmetric adversarial attack approach on TR decomposition (AdaTR) via min-max optimization, which can generate perturbation to the original tensor that significantly degrade the performance of TR decomposition. To address the high computational cost of AdaTR, we further propose a faster approximate gradient adversarial attack on TR decomposition (FAG-AdaTR) while maintaining strong attack effectiveness. Extensive experiments on color images, videos, and recommender systems demonstrate the effectiveness of the proposed methods in attacking TR decomposition and its applications. In future work, we will extend the proposed methods to other tensor decomposition models (Zheng et al., 2021; Wu et al., 2022; Loeschcke et al., 2024), and explore broader applications. In particular, when applied to large language model (LLM) compression (Hajimolahoseini et al., 2021; Ma et al., 2019), recommender systems (Chen et al., 2021), or tensor decomposition-based purification (Entezari & Papalexakis, 2022; Bhattarai et al., 2023), our approach highlights the importance of security issues in these domains, since their performance may also be affected by the vulnerability of tensor decomposition.

ETHICS STATEMENT

This work uses only computational methods and publicly available datasets, with no human subjects or private data. It follows the ICLR Code of Ethics, with no conflicts of interest. While acknowledging potential dual-use concerns, we stress responsible deployment and adhere to research integrity. All methods and results are reported transparently to support reproducibility.

REPRODUCIBILITY STATEMENT

We provide implementation details in the appendix to support reproduction of the main results.

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

## A  STATEMENT OF THE USE OF LARGE LANGUAGE MODELS (LLMS)

In this paper, we just used the LLM, ChatGPT, to polish the language of the paper. We did not use LLMs to generate any content or ideas in this work. We have verified the accuracy of all content and ideas in the paper.

## B  PROOF OF THEOREM 1

*Proof.* For standard TR-ALS, the tensor $\mathcal{X}$ can be expressed as

$$\mathcal{X} = \mathrm{TR}([\mathcal{G}]) + \mathcal{R}_1, \quad \|\mathcal{R}_1\|_{\mathrm{F}}^2 = \delta, \tag{20}$$

where the $\mathcal{R}_1$ is residual term of TR-ALS. For ATTR, we have

$$\mathcal{X} = \mathrm{TR}([\mathcal{G}]) - \mathcal{E} + \mathcal{R}_2, \quad \|-\mathcal{E} + \mathcal{R}_2\|_{\mathrm{F}}^2 \leq (\|\mathcal{E}\|_{\mathrm{F}} + \|\mathcal{R}_2\|_{\mathrm{F}})^2, \tag{21}$$

where $\|\mathcal{E}\|_{\mathrm{F}}^2 \leq \epsilon$.

If

$$(\|\mathcal{E}\|_{\mathrm{F}} + \|\mathcal{R}_2\|_{\mathrm{F}})^2 < \|\mathcal{R}_1\|_{\mathrm{F}}^2,$$

then ATTR yields a strictly smaller reconstruction error. Since $\|\mathcal{R}_1\|_{\mathrm{F}}^2 = \delta$ and $\|\mathcal{E}\|_{\mathrm{F}}^2 \leq \epsilon$, this condition is satisfied whenever

$$\sqrt{\epsilon} < \sqrt{\delta} - \|\mathcal{R}_2\|_{\mathrm{F}}.$$

Thus, ATTR achieves a smaller reconstruction error than standard TR-ALS under the stated condition. $\square$

## C  PROOF OF THEOREM 2

We first prove the smoothness of the surrogate objective, and boundedness of the variables. Then we prove they are the Cauchy sequence if Algorithm 1. To prove the boundedness of multipliers of Algorithm 1, we first introduce the following lemma.

### C.1  SMOOTHNESS OF THE SURROGATE OBJECTIVE

**Lemma 2.** *Chain rule for $\nabla_{\mathcal{E}} g(\mathcal{E})$ Assuming that: (i) the map $\mathcal{E} \mapsto [\mathcal{G}^{(T)}(\mathcal{E})]$ is differentiable on $\mathcal{B}$ with bounded Jacobian; (ii) the TR reconstruction $\mathrm{TR}(\cdot)$ is smooth on bounded sets.*

*Then, let*

$$h([\mathcal{G}]) = \tfrac{1}{2}\big\|\mathcal{X} - \mathrm{TR}\big([\mathcal{G}^{(T)}(\mathcal{E})]\big)\big\|_F^2. \tag{22}$$

*Then*

$$\nabla_{\mathcal{E}} g(\mathcal{E}) = J^{(T)}(\mathcal{E})^\top \nabla_{[\mathcal{G}^{(T)}(\mathcal{E})]} h\big([\mathcal{G}^{(T)}(\mathcal{E})]\big), \tag{23}$$

*where*

$$J^{(T)}(\mathcal{E}) = \frac{\partial \mathrm{vec}\big([\mathcal{G}^{(T)}(\mathcal{E})]\big)}{\partial \mathrm{vec}(\mathcal{E})}. \tag{24}$$

*Proof.* Let $e = \mathrm{vec}(\mathcal{E})$, $\theta(\mathcal{E}) = \mathrm{vec}([\mathcal{G}^{(T)}(\mathcal{E})])$. Then $g(\mathcal{E}) = h(\theta(\mathcal{E}))$. By the multivariate chain rule,

$$\nabla_e g = (\partial \theta / \partial e)^\top \nabla_\theta h(\theta),$$

giving Eq. (23).

$\square$

**Lemma 3.** *Lipschitz continuity of $\nabla_{\mathcal{E}} g(\mathcal{E})$*

*Assume that for all $\mathcal{E}_1, \mathcal{E}_2 \in \mathcal{B}$:*

$$\|[\mathcal{G}^{(T)}(\mathcal{E}_1)] - [\mathcal{G}^{(T)}(\mathcal{E}_2)]\|_F \leq L_G \|\mathcal{E}_1 - \mathcal{E}_2\|_F,$$

$$\|J^{(T)}(\mathcal{E})\| \le M_J, \qquad \|J^{(T)}(\mathcal{E}_1) - J^{(T)}(\mathcal{E}_2)\| \le L_J \|\mathcal{E}_1 - \mathcal{E}_2\|_F,$$

*and $\nabla_{[\mathcal{G}]} h([\mathcal{G}])$ is Lipschitz and bounded with constants $L_h, M_h$ on $\{[\mathcal{G}^{(T)}(\mathcal{E})] : \mathcal{E} \in \mathcal{B}\}$.*

*Then*

$$\|\nabla_{\mathcal{E}_1} g(\mathcal{E}_1) - \nabla_{\mathcal{E}_2} g(\mathcal{E}_2)\|_F \le L \|\mathcal{E}_1 - \mathcal{E}_2\|_F, \qquad L := M_J L_h L_G + L_J M_h. \tag{25}$$

*Proof.* Let

$$J_i := J^{(T)}(\mathcal{E}_i), \qquad [\mathcal{G}^{(T)}]_i := [\mathcal{G}^{(T)}(\mathcal{E}_i)], \qquad v_i := \nabla_{[\mathcal{G}^{(T)}]_i} h\left([\mathcal{G}^{(T)}]_i\right), \quad i = 1, 2. \tag{26}$$

By Lemma 1, one has

$$\nabla_{\mathcal{E}_i} g(\mathcal{E}_i) = J_i^\top v_i, \qquad i = 1, 2. \tag{27}$$

Thus,

$$\nabla_{\mathcal{E}_1} g(\mathcal{E}_1) - \nabla_{\mathcal{E}_2} g(\mathcal{E}_2) = J_1^\top (v_1 - v_2) + (J_1^\top - J_2^\top) v_2. \tag{28}$$

Taking norms and applying the triangle inequality yields

$$\left\|\nabla_{\mathcal{E}_1} g(\mathcal{E}_1) - \nabla_{\mathcal{E}_2} g(\mathcal{E}_2)\right\| \le \|J_1^\top (v_1 - v_2)\| + \|(J_1^\top - J_2^\top) v_2\| =: T_1 + T_2. \tag{29}$$

We bound the two terms separately.

BOUND ON $T_1$. By submultiplicativity of the operator norm,

$$T_1 = \|J_1^\top (v_1 - v_2)\| \le \|J_1^\top\| \, \|v_1 - v_2\| = \|J_1\| \, \|v_1 - v_2\|. \tag{30}$$

Using the assumption $\|J^{(T)}(\mathcal{E})\| \le M_J$,

$$\|J_1\| \le M_J. \tag{31}$$

Since $\nabla_{[\mathcal{G}]} h$ is $L_h$-Lipschitz,

$$\|v_1 - v_2\| = \left\|\nabla_{[\mathcal{G}^{(T)}]_1} h([\mathcal{G}^{(T)}]_1) - \nabla_{[\mathcal{G}^{(T)}]_2} h([\mathcal{G}^{(T)}]_2)\right\| \le L_h \left\|[\mathcal{G}^{(T)}]_1 - [\mathcal{G}^{(T)}]_2\right\|. \tag{32}$$

Finally, by the Lipschitz property of the ALS map,

$$\left\|[\mathcal{G}^{(T)}]_1 - [\mathcal{G}^{(T)}]_2\right\| \le L_G \|\mathcal{E}_1 - \mathcal{E}_2\|_F. \tag{33}$$

Combining Eq. (30)–Eq. (33) gives

$$T_1 \le M_J L_h L_G \|\mathcal{E}_1 - \mathcal{E}_2\|_F. \tag{34}$$

BOUND ON $T_2$. Similarly,

$$T_2 = \|(J_1^\top - J_2^\top) v_2\| \le \|J_1^\top - J_2^\top\| \, \|v_2\| = \|J_1 - J_2\| \, \|v_2\|. \tag{35}$$

Using the Lipschitz assumption on the $j_I$,

$$\|J_1 - J_2\| \le L_J \|\mathcal{E}_1 - \mathcal{E}_2\|_F. \tag{36}$$

Using the boundedness of $\nabla_{[\mathcal{G}]} h$,

$$\|v_2\| = \left\|\nabla_{[\mathcal{G}^{(T)}]_2} h([\mathcal{G}^{(T)}]_2)\right\| \le M_h. \tag{37}$$

Therefore,

$$T_2 \le L_J M_h \|\mathcal{E}_1 - \mathcal{E}_2\|_F. \tag{38}$$

FINAL BOUND. Combining Eq. (34) and Eq. (38) with Eq. (29) yields

$$\left\|\nabla_{\mathcal{E}_1} g(\mathcal{E}_1) - \nabla_{\mathcal{E}_2} g(\mathcal{E}_2)\right\| \le (M_J L_h L_G + L_J M_h) \|\mathcal{E}_1 - \mathcal{E}_2\|_F. \tag{39}$$

Thus the Lipschitz constant is $L = M_J L_h L_G + L_J M_h$, which proves Eq. (25).

$\square$

## C.2  PROOF OF THEOREM 2

*Proof.* Since $\nabla g$ is $L$-Lipschitz (Lemma 2), the function $g$ is $L$-smooth. Thus, for all $\mathcal{E}, \widetilde{\mathcal{E}} \in \mathcal{B}$,

$$g(\widetilde{\mathcal{E}}) \geq g(\mathcal{E}) + \langle \nabla g(\mathcal{E}), \widetilde{\mathcal{E}} - \mathcal{E} \rangle - \frac{L}{2} \|\widetilde{\mathcal{E}} - \mathcal{E}\|_F^2. \tag{40}$$

Let $\mathcal{E} = \mathcal{E}^{(t)}, \widetilde{\mathcal{E}} = \mathcal{E}^{(t+1)}$, and denote

$$\mathcal{Z}^t = \mathcal{E}^{(t)} + \eta_t \nabla g(\mathcal{E}^{(t)}).$$

Since $\mathcal{E}^{(t+1)} = \Pi_{\mathcal{B}}(\mathcal{Z}^t)$, the optimality condition of the Euclidean projection onto the convex set $\mathcal{B}$ implies

$$\langle \mathcal{Z}^t - \mathcal{E}^{(t+1)}, \mathcal{E}^{(t+1)} - \mathcal{E}^{(t)} \rangle \geq 0.$$

Substituting $\mathcal{Z}^t = \mathcal{E}^{(t)} + \eta_t \nabla g(\mathcal{E}^{(t)})$ yields

$$\eta_t \langle \nabla g(\mathcal{E}^{(t)}), \mathcal{E}^{(t+1)} - \mathcal{E}^{(t)} \rangle \geq \|\mathcal{E}^{(t+1)} - \mathcal{E}^{(t)}\|_F^2. \tag{41}$$

Plugging Eq. (41) into the smoothness inequality Eq. (40) gives

$$g(\mathcal{E}^{(t+1)}) \geq g(\mathcal{E}^{(t)}) + \left( \frac{1}{\eta_t} - \frac{L}{2} \right) \|\mathcal{E}^{(t+1)} - \mathcal{E}^{(t)}\|_F^2. \tag{42}$$

Since $\eta_t \leq 1/L$, the coefficient is nonnegative, proving monotonic ascent:

$$g(\mathcal{E}^{(t+1)}) \geq g(\mathcal{E}^{(t)}).$$

Because $\mathcal{B}$ is compact, $\{g(\mathcal{E}^{(t)})\}$ is monotone and bounded above, and hence convergent.

Summing Eq. (42) from $t = 0$ to $T$,

$$g(\mathcal{E}^{(t+1)}) - g(\mathcal{E}^0) \geq \sum_{t=0}^{T} \left( \frac{1}{\eta_t} - \frac{L}{2} \right) \|\mathcal{E}^{(t+1)} - \mathcal{E}^{(t)}\|_F^2.$$

Since $\eta_t \leq 1/L$,

$$\frac{1}{\eta_t} - \frac{L}{2} \geq \frac{L}{2} =: c > 0,$$

and because $g$ is bounded above on $\mathcal{B}$, letting $T \to \infty$ yields

$$\sum_{t=0}^{\infty} \|\mathcal{E}^{(t+1)} - \mathcal{E}^{(t)}\|_F^2 < \infty, \tag{43}$$

implying $\|\mathcal{E}^{(t+1)} - \mathcal{E}^{(t)}\|_F \to 0$. Hence, the sequence $\{\mathcal{E}^{(t)}\}$ is Cauchy sequence.

By compactness, $\{\mathcal{E}^{(t)}\}$ admits a convergent subsequence $\mathcal{E}^{t_k} \to \mathcal{E}^{\star}$. Since $\eta_t \in [\underline{\eta}, \bar{\eta}]$, we may assume $\eta_{t_k} \to \eta^{\star} \in [\underline{\eta}, \bar{\eta}]$. Using Eq. (43), $\mathcal{E}^{t_k+1} - \mathcal{E}^{t_k} \to 0$, hence $\mathcal{E}^{t_k+1} \to \mathcal{E}^{\star}$ as well.

Passing to the limit in the update rule,

$$\mathcal{E}^{t_k+1} = \Pi_{\mathcal{B}}\left( \mathcal{E}^{t_k} + \eta_{t_k} \nabla g(\mathcal{E}^{t_k}) \right),$$

and using continuity of $\nabla g$ and $\Pi_{\mathcal{B}}$, we obtain the fixed-point relation

$$\mathcal{E}^{\star} = \Pi_{\mathcal{B}}(\mathcal{E}^{\star} + \eta^{\star} \nabla g(\mathcal{E}^{\star})). \tag{44}$$

The projection fixed-point condition Eq. (44) is equivalent to the variational inequality

$$\langle \nabla g(\mathcal{E}^{\star}), \mathcal{Y} - \mathcal{E}^{\star} \rangle \leq 0, \qquad \forall \mathcal{Y} \in \mathcal{B},$$

which are exactly the KKT conditions for maximizing $g$ over $\mathcal{B}$. Thus, every limit point of the sequence is KKT-stationary.

$\square$

### C.3 PROOF OF LEMMA 1

*Proof.* Theorem 2 gives $g(\mathcal{E}^{(t+1)}) \geq g(\mathcal{E}^{(t)}) \geq g(\mathcal{E}^{(0)})$. Combining $g(\mathcal{E}^{(0)}) > g(\mathbf{0})$ with $g(\widetilde{\mathcal{E}}) < g(\mathbf{0})$ (Theorem 1) yields the claim. □

## D  PROOF OF THEOREM 3

In this appendix we establish the convergence of FAG-AdaTR stated in Theorem 3. The key observation is that each mode-wise loss $h_n(\mathbf{E}_{[n]})$ is a smooth quadratic function of $\mathbf{E}_{[n]}$, and thus its gradient is Lipschitz on bounded sets. Summing over $n$ preserves smoothness and Lipschitz continuity of the global surrogate $\tilde{g}$.

### D.1 SMOOTHNESS OF THE SURROGATE OBJECTIVE

We first show that every mode-wise loss $h_n$ has a Lipschitz continuous gradient.

**Lemma 4** (Lipschitz continuity of $\nabla h_n$). *Fix $n \in \{1, \dots, N\}$ and define*

$$\mathbf{M}_n := \mathbf{G}_{\neq n}^{(T-1)\,\dagger} \mathbf{G}_{\neq n}^{(T)} \in \mathbb{R}^{\prod_{j\neq n} I_j \times \prod_{j\neq n} I_j}. \tag{45}$$

*Assume that $\mathbf{M}_n$ is bounded on the perturbation ball $\mathcal{B}$, i.e., there exists $C_n > 0$ such that $\|\mathbf{M}_n\|_2 \leq C_n$ for all iterates. Then $h_n$ is $L_n$-smooth on $\mathcal{B}$ with*

$$L_n \;=\; \left\|\mathbf{M}_n^\top \mathbf{M}_n\right\|_2 \;\leq\; C_n^2. \tag{46}$$

*In particular, for any $\mathcal{E}_1, \mathcal{E}_2 \in B$,*

$$\left\|\nabla h_n((\mathbf{E}_1)_{[n]}) - \nabla h_n((\mathbf{E}_2)_{[n]})\right\|_F \;\leq\; L_n \left\|(\mathbf{E}_1)_{[n]} - (\mathbf{E}_2)_{[n]}\right\|_F. \tag{47}$$

*Proof.* By definition,

$$h_n(\mathbf{E}_{[n]}) = \frac{1}{2}\left\|\mathbf{X}_{[n]} - (\mathbf{X}_{[n]} + \mathbf{E}_{[n]})\mathbf{M}_n\right\|_F^2 = \frac{1}{2}\left\|\mathbf{R}_n - \mathbf{E}_{[n]}\mathbf{M}_n\right\|_F^2, \tag{48}$$

where $\mathbf{R}_n := \mathbf{X}_{[n]} - \mathbf{X}_{[n]}\mathbf{M}_n$ is independent of $\mathbf{E}_{[n]}$. Expanding the gradient of this quadratic form yields

$$\nabla_{\mathbf{E}_{[n]}} h_n(\mathbf{E}_{[n]}) = \left(\mathbf{E}_{[n]}\mathbf{M}_n - \mathbf{R}_n\right)\mathbf{M}_n^\top. \tag{49}$$

Thus, for any $(\mathbf{E}_1)_{[n]}, (\mathbf{E}_2)_{[n]}$,

$$\nabla h_n((\mathbf{E}_1)_{[n]}) - \nabla h_n((\mathbf{E}_2)_{[n]}) = \left(((\mathbf{E}_1)_{[n]} - (\mathbf{E}_2)_{[n]})\mathbf{M}_n\right)\mathbf{M}_n^\top, \tag{50}$$

$$\left\|\nabla h_n((\mathbf{E}_1)_{[n]}) - \nabla h_n((\mathbf{E}_2)_{[n]})\right\|_F \leq \left\|(\mathbf{E}_1)_{[n]} - (\mathbf{E}_2)_{[n]}\right\|_F \left\|\mathbf{M}_n\mathbf{M}_n^\top\right\|_2 \tag{51}$$

$$= \left\|\mathbf{M}_n^\top \mathbf{M}_n\right\|_2 \left\|(\mathbf{E}_1)_{[n]} - (\mathbf{E}_2)_{[n]}\right\|_F. \tag{52}$$

Therefore $L_n = \|\mathbf{M}_n^\top \mathbf{M}_n\|_2$ is a Lipschitz constant for $\nabla h_n$ on $\mathcal{B}$. The bound $L_n \leq C_n^2$ follows from $\|\mathbf{M}_n^\top \mathbf{M}_n\|_2 \leq \|\mathbf{M}_n\|_2^2$. □

We now lift this property from the mode-wise losses $h_n$ to the full surrogate $\tilde{g}(\mathbf{E}) = \sum_{n=1}^N \omega_n h_n(\mathbf{E}_{[n]})$.

**Lemma 5** (Lipschitz continuity of $\nabla \tilde{g}$). *Under the assumptions of Lemma 4, the surrogate objective $\tilde{g}$ is L-smooth on $\mathcal{B}$, with*

$$L \;\leq\; \sum_{n=1}^N \omega_n L_n. \tag{53}$$

*In particular, for all $\mathcal{E}_1, \mathcal{E}_2 \in B$,*

$$\left\|\nabla\tilde{g}(\mathcal{E}_1) - \nabla\tilde{g}(\mathcal{E}_2)\right\|_F \;\leq\; L \left\|\mathcal{E}_1 - \mathcal{E}_2\right\|_F. \tag{54}$$

*Proof.* By linearity of the gradient,

$$\nabla \tilde{g}(\mathcal{E}) = \sum_{n=1}^{N} \omega_n \operatorname{Fold}_n\big(\nabla h_n(\mathbf{E}_{[n]})\big), \tag{55}$$

where $\operatorname{Fold}_n(\cdot)$ denotes the inverse of the mode-$n$ unfolding. For any $\mathcal{E}_1, \mathcal{E}_2 \in B$,

$$\big\|\nabla \tilde{g}(\mathcal{E}_1) - \nabla \tilde{g}(\mathcal{E}_2)\big\|_F \leq \sum_{n=1}^{N} \omega_n \big\|\operatorname{Fold}_n\big(\nabla h_n((\mathbf{E}_1)_{[n]}) - \nabla h_n((\mathbf{E}_2)_{[n]})\big)\big\|_F \tag{56}$$

$$= \sum_{n=1}^{N} \omega_n \big\|\nabla h_n((\mathbf{E}_1)_{[n]}) - \nabla h_n((\mathbf{E}_2)_{[n]})\big\|_F \tag{57}$$

$$\leq \sum_{n=1}^{N} \omega_n L_n \big\|(\mathbf{E}_1)_{[n]} - (\mathbf{E}_2)_{[n]}\big\|_F \tag{58}$$

$$= \Big(\sum_{n=1}^{N} \omega_n L_n\Big) \|\mathcal{E}_1 - \mathcal{E}_2\|_F. \tag{59}$$

Thus $\nabla \tilde{g}$ is Lipschitz on $\mathcal{B}$ with constant $L \leq \sum_{n=1}^{N} \omega_n L_n$. $\qquad\square$

### D.2 PROOF OF THEOREM 3

*Proof.* By Lemma 5, $\tilde{g}$ is $L$-smooth on the compact set $\mathcal{B}$. For any $\mathcal{E}, \mathcal{E}' \in B$, $L$-smoothness implies the standard inequality

$$\tilde{g}(\mathcal{E}') \geq \tilde{g}(\mathcal{E}) + \langle \nabla \tilde{g}(\mathcal{E}), \mathcal{E}' - \mathcal{E} \rangle - \frac{L}{2}\|\mathcal{E}' - \mathcal{E}\|_F^2. \tag{60}$$

Let $\mathcal{E} = \mathcal{E}^{(t)}$ and $\mathcal{Z}^{(t)} = \mathcal{E}^{(t)} + \eta_t \nabla \tilde{g}(\mathcal{E}^{(t)})$. By the optimality condition of the Euclidean projection onto the convex set $\mathcal{B}$, the update $\mathcal{E}^{(t+1)} = \Pi_B(\mathcal{Z}^{(t)})$ satisfies

$$\big\langle \mathcal{Z}^{(t)} - \mathcal{E}^{(t+1)}, \mathcal{E}^{(t+1)} - \mathcal{E}^{(t)} \big\rangle \geq 0. \tag{61}$$

Substituting $\mathcal{Z}^{(t)}$ and rearranging yields

$$\eta_t \big\langle \nabla \tilde{g}(\mathcal{E}^{(t)}), \mathcal{E}^{(t+1)} - \mathcal{E}^{(t)} \big\rangle \geq \|\mathcal{E}^{(t+1)} - \mathcal{E}^{(t)}\|_F^2. \tag{62}$$

Combining with $L$-smoothness (with $\mathcal{E}' = \mathcal{E}^{(t+1)}$) gives

$$\tilde{g}(\mathcal{E}^{(t+1)}) \geq \tilde{g}(\mathcal{E}^{(t)}) + \Big(\frac{1}{\eta_t} - \frac{L}{2}\Big)\|\mathcal{E}^{(t+1)} - \mathcal{E}^{(t)}\|_F^2. \tag{63}$$

By the step size condition $\eta_t \leq 1/L$, the coefficient $\frac{1}{\eta_t} - \frac{L}{2}$ is nonnegative, and thus $\tilde{g}(\mathcal{E}^{(t+1)}) \geq \tilde{g}(\mathcal{E}^{(t)})$ for all $t$. Since $\mathcal{B}$ is compact and $\tilde{g}$ is continuous, $\tilde{g}$ is bounded above on $\mathcal{B}$, so the monotone sequence $\{\tilde{g}(\mathcal{E}^{(t)})\}$ converges.

Summing the inequality over $t = 0, \ldots, T$ and using $\eta_t \leq 1/L$ yields

$$\sum_{t=0}^{T} \|\mathcal{E}^{(t+1)} - \mathcal{E}^{(t)}\|_F^2 \leq \frac{2}{L}\big(\tilde{g}(\mathcal{E}^{(T+1)}) - \tilde{g}(\mathcal{E}^{(0)})\big) \leq \frac{2}{L}\big(\sup_{\mathcal{E} \in B} \tilde{g}(\mathcal{E}) - \tilde{g}(\mathcal{E}^{(0)})\big) < \infty. \tag{64}$$

Hence $\|\mathcal{E}^{(t+1)} - \mathcal{E}^{(t)}\|_F \to 0$ and $\{\mathcal{E}^{(t)}\}$ is a Cauchy sequence in the complete metric space $\mathcal{B}$, so it converges.

Finally, let $\mathcal{E}^\star$ be any limit point of $\{\mathcal{E}^{(t)}\}$ and consider a subsequence $\mathcal{E}^{(t_k)} \to \mathcal{E}^\star$. Since $\|\mathcal{E}^{(t_k+1)} - \mathcal{E}^{(t_k)}\|_F \to 0$, we also have $\mathcal{E}^{(t_k+1)} \to \mathcal{E}^\star$. Passing to the limit in

$$\mathcal{E}^{(t_k+1)} = \Pi_B\big(\mathcal{E}^{(t_k)} + \eta_{t_k} \nabla \tilde{g}(\mathcal{E}^{(t_k)})\big) \tag{65}$$

and using continuity of $\Pi_B$ and $\nabla \tilde{g}$ yields the fixed-point relation

$$\mathcal{E}^\star = \Pi_B\big(\mathcal{E}^\star + \eta^\star \nabla \tilde{g}(\mathcal{E}^\star)\big), \tag{66}$$

for some accumulation point $\eta^\star \in [\eta, \bar{\eta}]$. This fixed-point condition is equivalent to the first-order optimality (KKT stationarity) condition for the constrained maximization $\max_{\mathcal{E} \in B} \tilde{g}(\mathcal{E})$, namely

$$\langle \nabla \tilde{g}(\mathcal{E}^\star), \mathcal{Y} - \mathcal{E}^\star \rangle \leq 0, \quad \forall \mathcal{Y} \in B. \tag{67}$$

Thus every limit point of $\{\mathcal{E}^{(t)}\}$ is a KKT point of $\max_{\mathcal{E} \in B} \tilde{g}(\mathcal{E})$, which completes the proof. $\quad\square$

# E    PROOF OF THEOREM 4

Let $\mathcal{E}^\star$ be any limit point of FAG-AdaTR. From Appendix D, every such limit point satisfies the KKT stationarity condition for the surrogate maximization problem $\max_{\mathcal{E} \in B} \tilde{g}(\mathcal{E})$:

$$\langle \nabla \tilde{g}(\mathcal{E}^\star), \mathcal{Y} - \mathcal{E}^\star \rangle \leq 0, \qquad \forall \mathcal{Y} \in B, \tag{68}$$

where

$$B := \{ \mathcal{E} : \|\mathcal{E}\|_F^2 \leq \epsilon \}.$$

For any $\mathcal{Y} \in B$, decompose

$$\langle \nabla g(\mathcal{E}^\star), \mathcal{Y} - \mathcal{E}^\star \rangle = \langle \nabla \tilde{g}(\mathcal{E}^\star), \mathcal{Y} - \mathcal{E}^\star \rangle + \langle \nabla g(\mathcal{E}^\star) - \nabla \tilde{g}(\mathcal{E}^\star), \mathcal{Y} - \mathcal{E}^\star \rangle. \tag{69}$$

The first term is nonpositive due to Eq. (68). For the second term, apply the Cauchy–Schwarz inequality together with the gradient mismatch bound $\|\nabla g(\mathcal{E}) - \nabla \tilde{g}(\mathcal{E})\|_F \leq \varepsilon_g$:

$$\left| \langle \nabla g(\mathcal{E}^\star) - \nabla \tilde{g}(\mathcal{E}^\star), \mathcal{Y} - \mathcal{E}^\star \rangle \right| \leq \varepsilon_g \left\| \mathcal{Y} - \mathcal{E}^\star \right\|_F. \tag{70}$$

Because both $\mathcal{Y}$ and $\mathcal{E}^\star$ lie in the radius-$\sqrt{\epsilon}$ Frobenius ball $\mathcal{B}$, we have

$$\left\| \mathcal{Y} - \mathcal{E}^\star \right\|_F \leq \left\| \mathcal{Y} \right\|_F + \left\| \mathcal{E}^\star \right\|_F \leq 2\sqrt{\epsilon}.$$

Hence,

$$\left| \langle \nabla g(\mathcal{E}^\star) - \nabla \tilde{g}(\mathcal{E}^\star), \mathcal{Y} - \mathcal{E}^\star \rangle \right| \leq 2\sqrt{\epsilon}\, \varepsilon_g. \tag{71}$$

Combining the bounds for the two terms yields

$$\langle \nabla g(\mathcal{E}^\star), \mathcal{Y} - \mathcal{E}^\star \rangle \geq -2\sqrt{\epsilon}\, \varepsilon_g, \qquad \forall \mathcal{Y} \in B. \tag{72}$$

This shows that $\mathcal{E}^\star$ is an $O(\sqrt{\epsilon}\, \varepsilon_g)$-approximate stationary point of $g(\mathcal{E})$, completing the proof.

# F    ALGORITHMIC DETAILS

Here we provide the detailed pseudocode for the proposed methods.

---

**Algorithm 1** Adversarial Attack on TR Decomposition (AdaTR)

---

1: **Input:** tensor $\mathcal{X}$, attack budget $\epsilon$, learning rate $\eta$, outer iterations $T_{\text{out}}$, inner iterations $T_{\text{in}}$, TR ranks $R$
2: **Output:** adversarial tensor $\hat{\mathcal{X}}$
3: Initialize perturbation $\mathcal{E} \sim \mathcal{N}(0, 1)$ and project to $\|\mathcal{E}\|_F^2 \leq \epsilon$
4: **for** $t = 1$ to $T_{\text{out}}$ **do**
5:     $\mathcal{E}_{\text{old}} \leftarrow \mathcal{E}$
6:     $\hat{\mathcal{X}} \leftarrow \mathcal{X} + \mathcal{E}$
7:     **for** $k = 1$ to $T_{\text{in}}$ **do**
8:         Update factors $[\mathcal{G}]$ via Eq. (8) on $\hat{\mathcal{X}}$
9:     **end for**
10:    Compute loss $L = -g = -\|\mathcal{X} - \text{TR}([\mathcal{G}^{(T)}(\mathcal{E})])\|_F^2$
11:    Update $\mathcal{E} \leftarrow \mathcal{E} + \eta \frac{\partial g}{\partial \mathcal{E}}$    (backpropagation)
12:    Project $\mathcal{E}$ to $\|\mathcal{E}\|_F^2 \leq \epsilon$
13:    If $\|\mathcal{E} - \mathcal{E}_{\text{old}}\|_F / \|\mathcal{E}_{\text{old}}\|_F < \text{tol}$, break
14: **end for**
15: Return $\hat{\mathcal{X}} = \mathcal{X} + \mathcal{E}$

---

# G    COMPLEXITY ANALYSIS

Assuming the TR rank $R_1 = \cdots = R_N = R$ and data size $I_1 = \cdots = I_N = I$, the time complexity of one TR-ALS inner iteration is $\mathcal{O}(NI^N R^4 + NR^6)$ (He & Atia, 2023). AdaTR performs $T_{\text{in}}$ times inner iterations, and the backward pass has the same order as the forward computation, so each outer iteration costs $\mathcal{O}(2T_{\text{in}}(NI^N R^4 + NR^6))$. FAG-AdaTR uses the closed-form gradient instead of back-propagation through TR-ALS, and thus each outer iteration still costs $\mathcal{O}(T_{\text{in}}(NI^N R^4 + NR^6))$, but with a smaller constant factor in practice.

**Algorithm 2** Faster Approximate Gradient Adversarial Attack on TR Decomposition (FAG-AdaTR)

---

1: **Input:** tensor $\mathcal{X}$, attack budget $\epsilon$, learning rate $\eta$, outer iterations $T_{\text{out}}$, inner iterations $T_{\text{in}}$, TR ranks $R$
2: **Output:** adversarial tensor $\hat{\mathcal{X}}$
3: Initialize perturbation $\mathcal{E} \sim \mathcal{N}(0, 1)$ and project to $\|\mathcal{E}\|_{\text{F}} \leq \epsilon$
4: **for** $t = 1$ to $T_{\text{out}}$ **do**
5:    $\mathcal{E}_{\text{old}} \leftarrow \mathcal{E}$
6:    $\hat{\mathcal{X}} \leftarrow \mathcal{X} + \mathcal{E}$
7:    **for** $k = 1$ to $T_{\text{in}}$ **do**
8:       Update TR factors $[\mathcal{G}]$ by Eq. (8) on $\hat{\mathcal{X}}$
9:    **end for**
10:   Update the gradient of $\nabla_{\mathbf{E}_{[n]}} h_n(\mathbf{E}_{[n]})$ by Eq. (16)
11:   Update perturbation $\mathcal{E}$ by Eq. (17) with $[\mathcal{G}]$
12:   Project $\mathcal{E}$ to $\|\mathcal{E}\|_{\text{F}}^2 \leq \epsilon$
13:   If $\|\mathcal{E} - \mathcal{E}_{\text{old}}\|_{\text{F}} / \|\mathcal{E}_{\text{old}}\|_{\text{F}} < $ tol, break
14: **end for**
15: Return $\hat{\mathcal{X}} = \mathcal{X} + \mathcal{E}$

---

# H ADDITIONAL EXPERIMENTAL

## H.1 BASELINE METHODS

The attack methods used in this paper are:

- *Gaussian Noise*: The budget of Gaussian noise is consistent with the other methods, which satisfy $\|\mathcal{E}\|_{\text{F}}^2 \leq \epsilon$. We add Gaussian noise to the clean tensor $\mathcal{X}$ to get the adversarial tensor.

- *ATTR-gen*: We adopt the ATTR formulation $\max_{\|\mathcal{E}\|_{\text{F}}^2 \leq \epsilon} \min_{[\mathcal{G}]} \frac{1}{2} \|\mathcal{X} + \mathcal{E} - \text{TR}([\mathcal{G}])\|_{\text{F}}^2$ with the same perturbation budget $\epsilon$ as the other baselines.

- *AdaTR-gen*: The proposed AdaTR method generates the adversarial tensor under the given perturbation budget.

- *FAG-AdaTR-gen*: The proposed FAG-daTR method generates the adversarial tensor under the given perturbation budget.

The defense methods used in this paper are:

- *TR-ALS* (Zhao et al., 2016): The target model to evaluate various adversarial attack algorithms.

- *TRPCA-TNN* (Lu et al., 2019): This method aims to recover the low-rank and sparse tensor from the original tensor, which might defend against adversarial attacks on the tensor ring decomposition.

- *TRNNM* (Yu et al., 2019): This method completes tensors by enforcing a nuclear norm under the tensor ring structure, which may help suppress adversarial perturbations.

- *HQTRC* (He & Atia, 2022): This method leverages the coarse-to-fine framework to improve the robustness of the tensor ring decomposition.

- *LRTC-TV* (Li et al., 2017): This method uses the local smooth and piecewise priors to improve the recovery accuracy.

## H.2 IMPLEMENTATION DETAILS

We provide the detailed parameter settings and implementation environment used in our experiments.

The peak signal-to-noise rate (PSNR), the structural similarity (SSIM) (Wang et al., 2004), and residual standard error (RSE) are three quality metrics we used for numerical comparison. Besides, the hyperparameters of comparison algorithms are fine-tuned to the best results according to the

suggested range given by the authors in all the following experiments. For all methods, the maximum numbers of both inner and outer iterations were fixed at 100 by default. The TR-rank is fixed to $R_1 = R_2 = R_3 = 5$ throughout all experiments; if a different rank is used, it will be explicitly stated. Moreover, the values of $\epsilon$ and $\eta$ are set to 500 and 0.01, respectively by default. It is worth noting that all experiments use random sampling obeying a uniform distribution. We implement these algorithms on a remote server running Ubuntu 20.04 LTS with 256 GB RAM and a single NVIDIA RTX A5000 GPU (24 GB).

### H.3 RUNNING TIME ANALYSIS

To further compare the computational efficiency of AdaTR and FAG-AdaTR, we report their average running time and variance across color videos and color images under the same size of input tensor, rank, and inner ALS iterations. The results are summarized in Table 4, showing that FAG-AdaTR achieves a significant speedup over AdaTR while maintaining comparable attack effectiveness.

Table 4: Comparison of FAG-AdaTR and AdaTR in runtime and peak memory on color videos and color images. Runtime is reported as mean $\pm$ variance (seconds).

| Method | Video Time | Image Time | Video Memory | Image Memory |
|---|---|---|---|---|
| FAG-AdaTR | $28.55 \pm 4.93$ | $34.57 \pm 0.33$ | 999.82 MB | 50.04 MB |
| AdaTR | $44.78 \pm 12.91$ | $53.62 \pm 0.56$ | 8.57 GB | 531.39 MB |

### H.4 EXTENTION TO OTHER TENSOR TECOMPOSITIONS

In this section, we extend our experiments to Tucker-ALS, CP-ALS, and TT-ALS by directly replacing the TR decomposition operator in our asymmetric bilevel objective with the corresponding multilinear operators. All of these experiments are tested in the 8 color images from the DIV2K dataset. Due to the Ada-Tucker and Ada-CP requiring more computation resources, which might cause CUDA to run out of memory, we resize the image to the same small size of $150 \times 150 \times 3$ for all experiments.

Table 5: Comparison of reconstruction error under clean and attack conditions.

| Method | Clean Mean$\pm$Std | Attack Mean$\pm$Std |
|---|---|---|
| TR | 11.714$\pm$5.108 | **19.209$\pm$4.080** |
| TT | 11.998$\pm$4.996 | **19.633$\pm$4.037** |
| Tucker | 12.704$\pm$5.296 | **19.650$\pm$3.957** |
| CP | 12.310$\pm$5.223 | **17.458$\pm$3.868** |

The results are summarized in Table 5, showing that the proposed attack framework is effective across different tensor decomposition methods.

### H.5 EFFECTIVENESS OF ATTR AS A DEFENSE

To further examine the defensive potential of ATTR, we evaluate its performance under different perturbation budgets. Table 6 summarizes the results for $\epsilon = 10$ and $\epsilon = 100$ on the image decomposition task.

When the perturbation budget is small (e.g., $\epsilon = 10$), ATTR shows a limited defensive effect. However, as the perturbation budget increases (e.g., $\epsilon = 100$), the defensive effect becomes negligible. Both reconstruction error (RSE) and perceptual metrics (PSNR/SSIM) deteriorate significantly, and ATTR fails to prevent the attack from degrading performance.

In summary, while ATTR can provide marginal robustness under small perturbations, it does not offer effective defense against stronger adversarial attacks.

Table 6: Average performance on 8 images under different attacks. Metrics: PSNR (↑), RSE (↓), SSIM (↑). Best attack (largest RSE, lowest PSNR/SSIM) is highlighted in **bold**.

| Method | ATTR | | | TRALS | | |
|---|---|---|---|---|---|---|
| | PSNR | RSE | SSIM | PSNR | RSE | SSIM |
| $\epsilon = 10$ | | | | | | |
| Clean | 22.0015 | 0.182255 | 0.566261 | 21.9238 | 0.183665 | 0.563948 |
| Gaussian Noise | 22.0017 | 0.182268 | 0.566273 | 21.9238 | 0.183665 | 0.563933 |
| FAG-AdaTR-gen | 21.9997 | 0.182327 | 0.566151 | 21.9210 | 0.183741 | 0.563788 |
| ATTR-gen | **21.9929** | **0.182361** | **0.565766** | **21.8366** | **0.184679** | **0.561431** |
| $\epsilon = 100$ | | | | | | |
| Clean | 22.0006 | 0.182191 | 0.565995 | 21.8897 | 0.184207 | 0.562359 |
| Gaussian Noise | 21.8999 | 0.183687 | 0.546379 | 21.7429 | 0.185253 | 0.541178 |
| FAG-AdaTR-gen | 19.0929 | 0.236054 | 0.469656 | 19.0619 | 0.236838 | 0.467986 |
| ATTR-gen | **18.3931** | **0.263973** | **0.287658** | **18.2523** | **0.267261** | **0.281135** |

Table 7: Performance on MovieLens-100 under $F$-norm and $L_\infty$-norm perturbations with 10% sample ratios. Three evaluation metrics are reported: RMSE, Precision@10 (P@10), and Recall@10 (R@10).

| Method | RMSE | | P@10 | | R@10 | |
|---|---|---|---|---|---|---|
| | $F$ | $L_\infty$ | $F$ | $L_\infty$ | $F$ | $L_\infty$ |
| ATNMF | **4.025488** | **4.026136** | 0.012267 | 0.011200 | 0.043085 | 0.043204 |
| AdaNMF | 4.023416 | 4.025945 | **0.007733** | **0.005867** | **0.026103** | **0.023211** |

## H.6 EVALUATION UNDER $F$-NORM AND $L_\infty$-NORM PERTURBATIONS

In this section, we propose $L_\infty$-Norm perturbations to evaluate the performance of AdaTR and FAG-AdaTR on the MovieLens-100 dataset, 8 color images, and 7 videos, in addition to the previously used $F$-Norm perturbations. The results are summarized in Table 7 and Table 8. Noting that $L_\infty$-Norm perturbations and $F_\infty$-Norm perturbations have the same energy budget.

Overall, these results show that the proposed attack remains the most effective under both $F$-norm and $L_\infty$-norm constraints. For visual data such as images and videos, $F$-norm perturbations may create sharp local artifacts that are perceptible to humans, so the $L_\infty$ constraint is generally preferable and achieves comparable attack strength without noticeable distortions. In contrast, recommendation models operate on latent tensors that are not directly observed, allowing $F$-norm attacks to exploit a larger feasible space and thus yield stronger perturbations. Accordingly, we recommend using $L_\infty$ for perceptual data and $F$-norm for recommendation tasks.

## H.7 ADDITIONAL EXPERIMENT ON PRINCIPAL ANGLE MAXIMIZATION ATTACK

In this section, we provide an additional experiment that evaluates our method under a recently–considered adversarial strategy based on *principal angle maximization*. This attack is inspired by classical subspace-based adversarial analysis, where the adversary seeks a rank-one perturbation $\Delta\mathbf{X} = \mathbf{a}\mathbf{b}^\top$ that maximizes the largest principal angle between the clean feature subspace and the perturbed feature subspace. Such an attack is related to the method proposed in Li et al. (2020) and aims at shifting the principal components as much as possible within a constrained perturbation budget.

We follow this principal-angle attack formulation and apply it to our vision reconstruction setting. Six images are randomly sampled from the evaluation set, and each image is perturbed by the sub-

Table 8: Performance on Video and Image datasets under clean, $F$-norm, and $L_\infty$-norm perturbations. Best (strongest) attack results are highlighted.

| Method | Video (Reconstruction Error) | | | Image (Reconstruction Error) | | |
|---|---|---|---|---|---|---|
| | Clean | $F$ Attack | $L_\infty$ Attack | Clean | $F$ Attack | $L_\infty$ Attack |
| AdaTR | 145.9135 | **193.9599** | **188.0390** | 11.5323 | **23.4008** | **22.3264** |
| FAG-AdaTR | 145.9135 | 184.8151 | 162.4317 | 11.5323 | 22.1166 | 20.4082 |
| ATTR | 145.9135 | 156.9865 | 156.9865 | 11.5323 | 17.0802 | 17.0802 |

space attack under the same perturbation budget as in our main experiments. All the TR ranks are selected as 3.

For each method, we report the reconstruction error. The results are summarized in Table 9.

Table 9: Reconstruction errors under principal angle maximization attack on six randomly selected images. Lower is better.

| Image | Clean | Gaussian | ATTR | PCA_Attack | FAG-AdaTR | AdaTR |
|---|---|---|---|---|---|---|
| 1 | 7.4594 | 7.5276 | 7.4819 | 7.7084 | 8.4119 | **9.2618** |
| 2 | 7.7996 | 7.8555 | 7.9134 | 7.9134 | 8.6990 | **10.3844** |
| 3 | 10.3183 | 10.3592 | 10.2125 | 10.4432 | 11.0245 | **12.6237** |
| 4 | 7.5878 | 7.6523 | 7.6144 | 7.8395 | 8.5204 | **9.1112** |
| 5 | 16.8553 | 16.8838 | 16.8432 | 16.9065 | 17.2952 | **18.4781** |
| 6 | 14.0037 | 14.0385 | 13.7092 | 14.0992 | 14.5313 | **15.5617** |

**Discussion.** Although the principal-angle maximization attack is effective in shifting the underlying subspace, our methods still significantly outperform it. The reason is that the subspace deviation induced by principal-angle maximization does *not* necessarily correspond to a large reconstruction error. Principal angles measure the worst-case discrepancy between subspaces, but they do not directly control how the corrupted features affect pixel-level reconstruction. In contrast, both AdaTR and FAG-AdaTR are designed to minimize the *actual reconstruction error*, and thus remain robust even when the adversary succeeds in enlarging the principal angle.

## H.8 ADDITIONAL EXPERIMENTAL RESULTS

We include the complete experimental results of PSNR and RSE in color images decomposition (Tab. 10 and Tab. 11), visual results in color images decomposition (Fig. 7-14), visual results in color video decomposition (Fig. 15-21), and visual results in tensor completion (Fig. 22-27) in the appendix due to the space limitation of the paper.

Table 10: PSNR matrix: mean $\pm$ variance across runs. Higher is better; **bold** marks the worst (lowest) PSNR) per defense.

| Attack \ Defense | TR-ALS | TRPCA-TNN | TRNNM | HQTRC-Cor | HQTRC-Cau | HQTRC-Hub | LRTC-TV |
|---|---|---|---|---|---|---|---|
| Clean | 20.808 ± 18.572 | 26.349 ± 12.237 | 23.813 ± 4.898 | 22.864 ± 24.112 | 23.242 ± 21.633 | 25.327 ± 18.472 | 23.165 ± 9.119 |
| Gauss Noise | 19.539 ± 8.054 | 14.689 ± 0.302 | 11.485 ± 0.031 | 15.459 ± 0.152 | 15.856 ± 0.234 | 14.731 ± 0.091 | 17.619 ± 0.384 |
| ATTR-gen | 17.734 ± 11.846 | 11.315 ± 0.016 | 11.286 ± 0.027 | 15.294 ± 0.134 | 15.420 ± 0.122 | 15.071 ± 0.128 | 16.668 ± 0.991 |
| AdaTR-gen | **8.547 ± 0.388** | 9.892 ± 0.327 | 9.877 ± 0.061 | 13.477 ± 0.784 | 13.662 ± 1.063 | 13.195 ± 0.342 | 15.902 ± 0.550 |
| FAG-AdaTR-gen | 8.800 ± 0.106 | **9.314 ± 0.026** | **9.516 ± 0.016** | **11.357 ± 1.576** | **11.492 ± 1.712** | **11.125 ± 0.914** | **11.592 ± 1.087** |

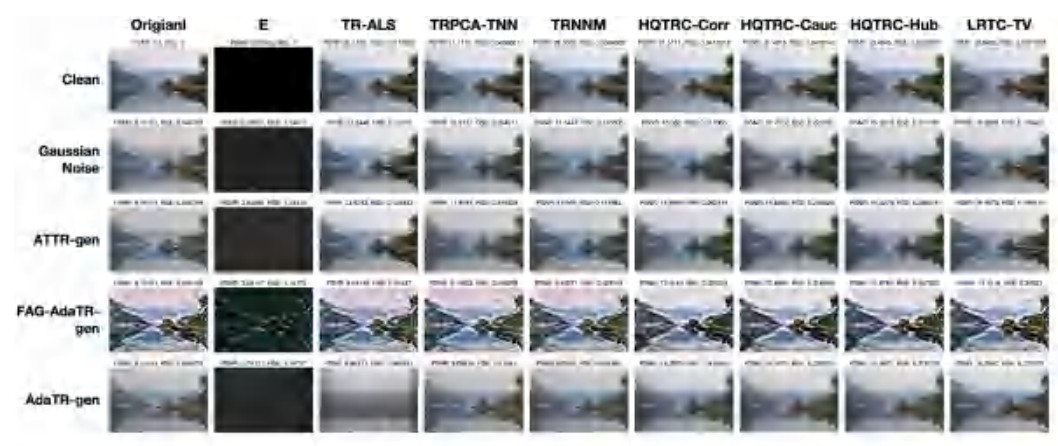

Figure 7: Visual results on tensor decomposition tasks under different attacks and defenses for Image 1.

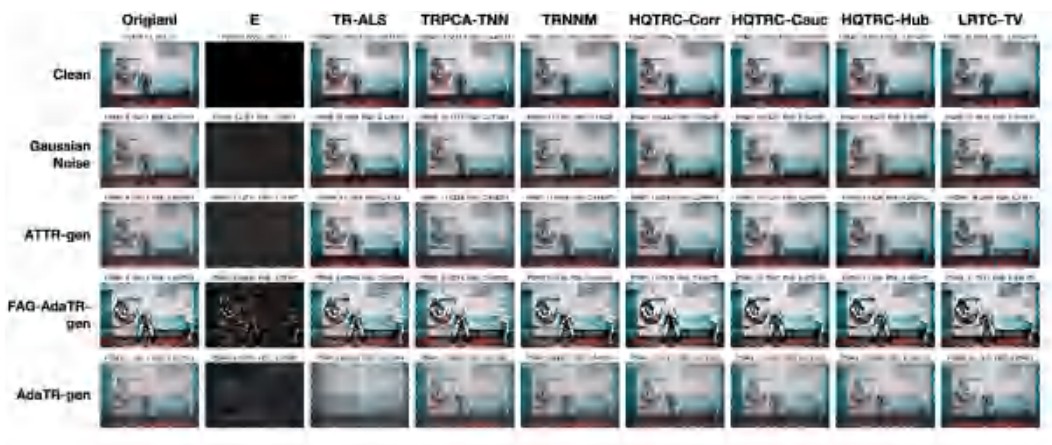

Figure 8: Visual results on tensor decomposition tasks under different attacks and defenses for Image 2.

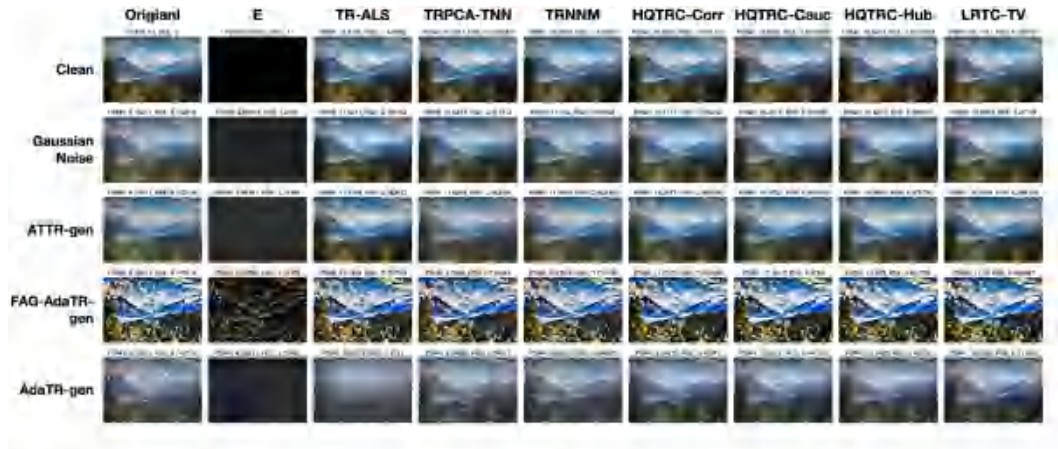

Figure 9: Visual results on tensor decomposition tasks under different attacks and defenses for Image 3.

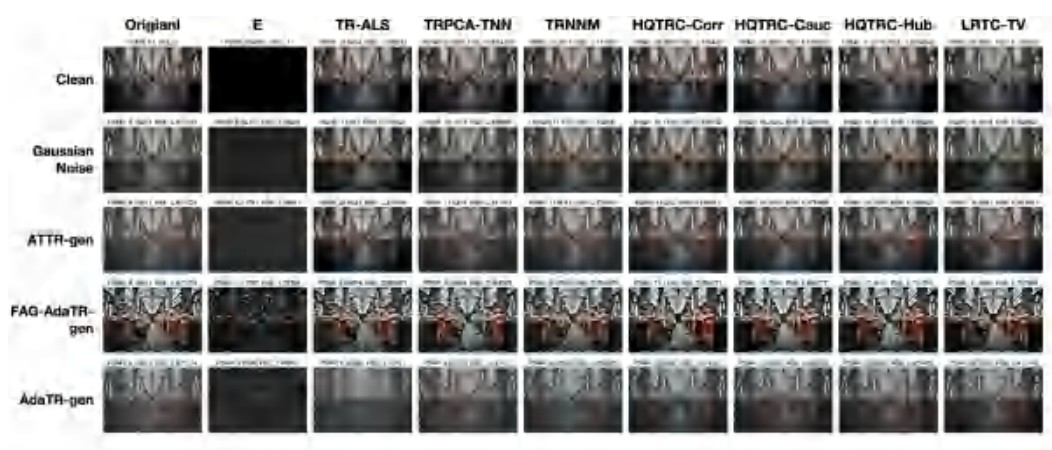

Figure 10: Visual results on tensor decomposition tasks under different attacks and defenses for Image 4.

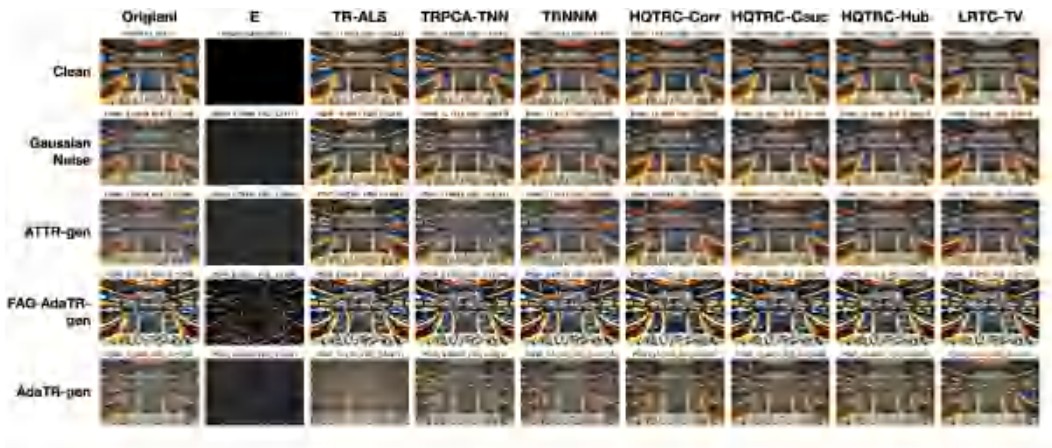

Figure 11: Visual results on tensor decomposition tasks under different attacks and defenses for Image 5.

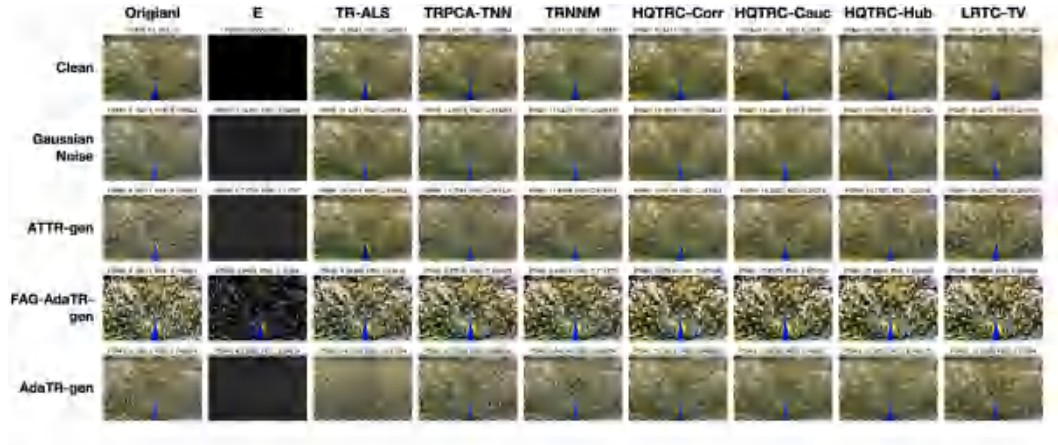

Figure 12: Visual results on tensor decomposition tasks under different attacks and defenses for Image 6.

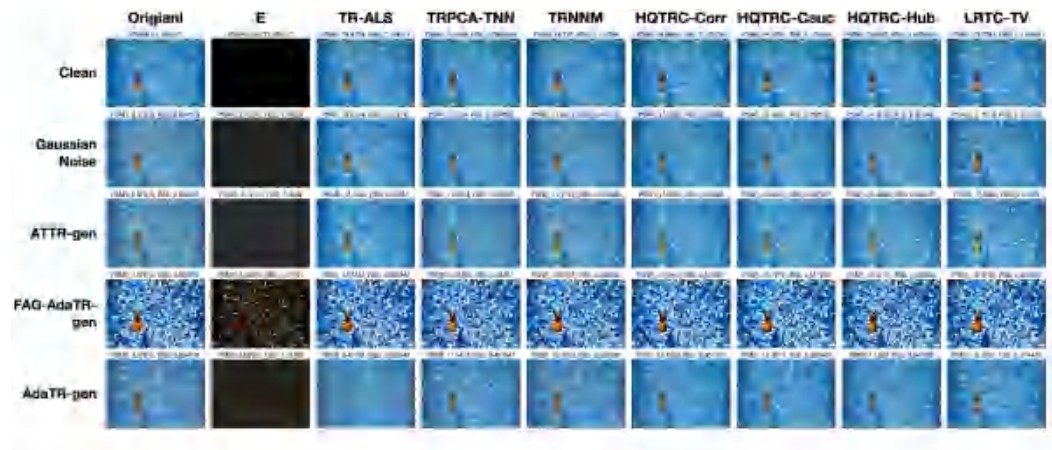

Figure 13: Visual results on tensor decomposition tasks under different attacks and defenses for Image 7.

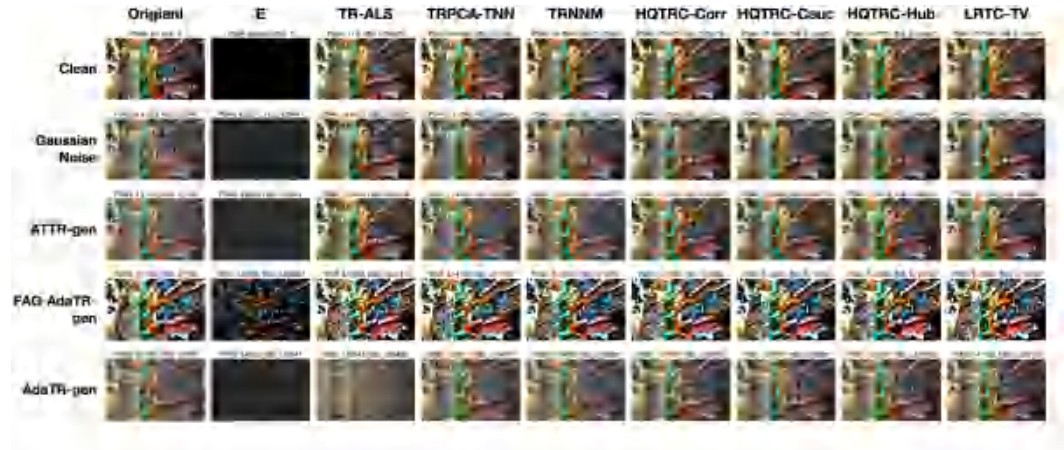

Figure 14: Visual results on tensor decomposition tasks under different attacks and defenses for Image 8.

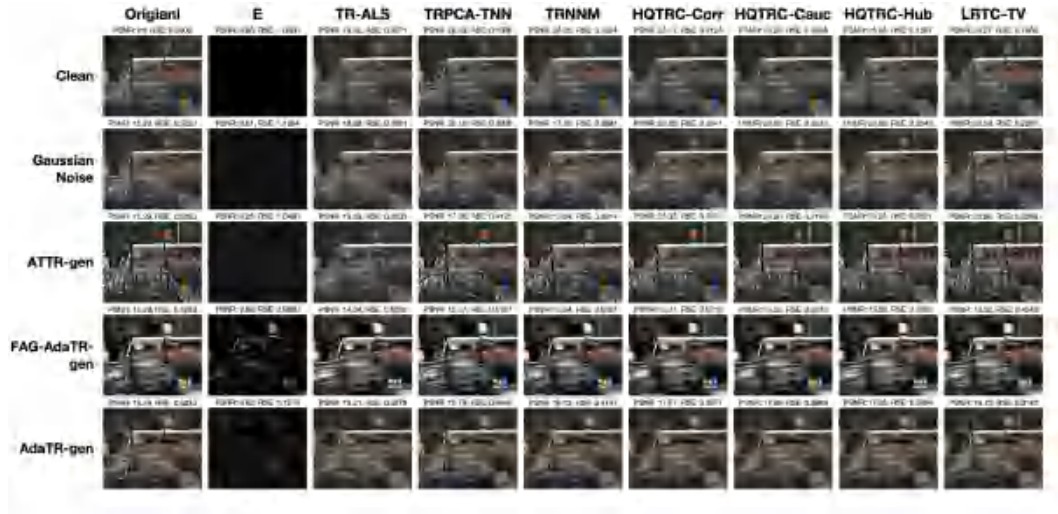

Figure 15: Visual results on tensor decomposition tasks under different attacks and defenses for the 5th frame of Video 1.

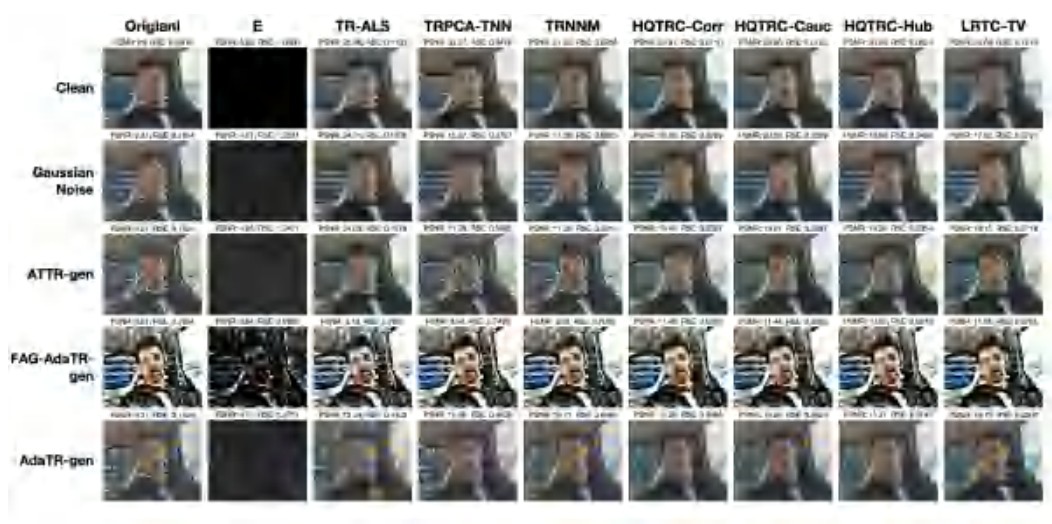

Figure 16: Visual results on tensor decomposition tasks under different attacks and defenses for the 5th frame of Video 2.

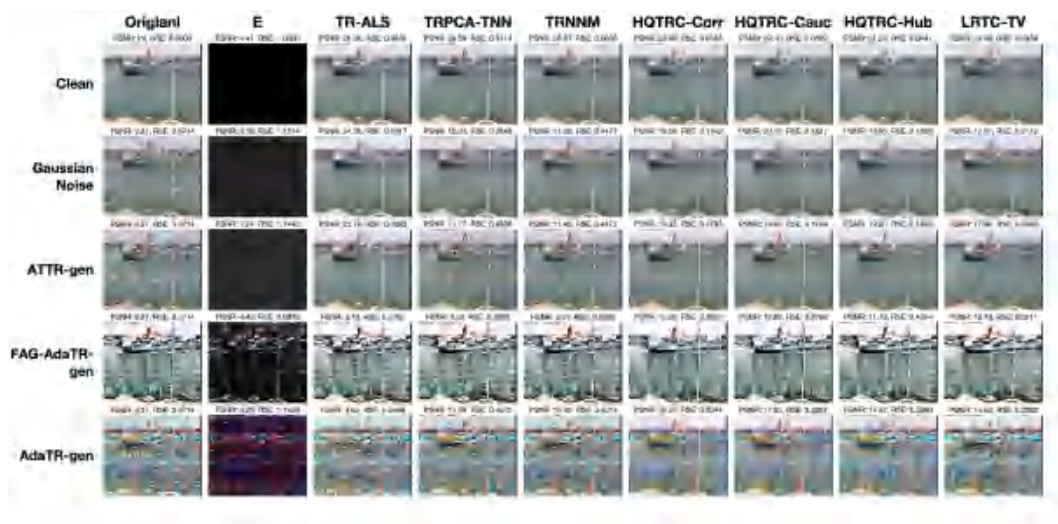

Figure 17: Visual results on tensor decomposition tasks under different attacks and defenses for the 5th frame of Video 3.

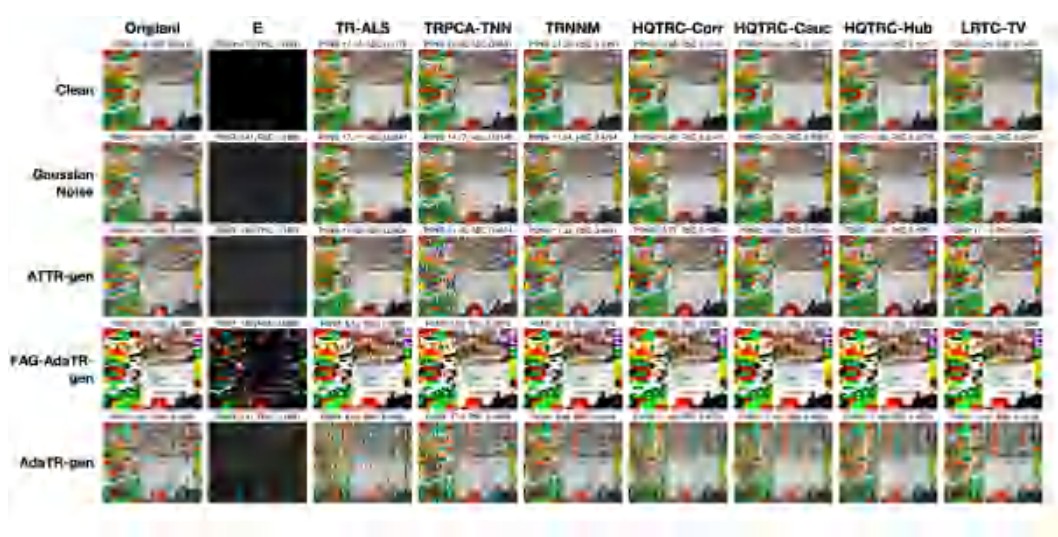

Figure 18: Visual results on tensor decomposition tasks under different attacks and defenses for the 5th frame of Video 4.

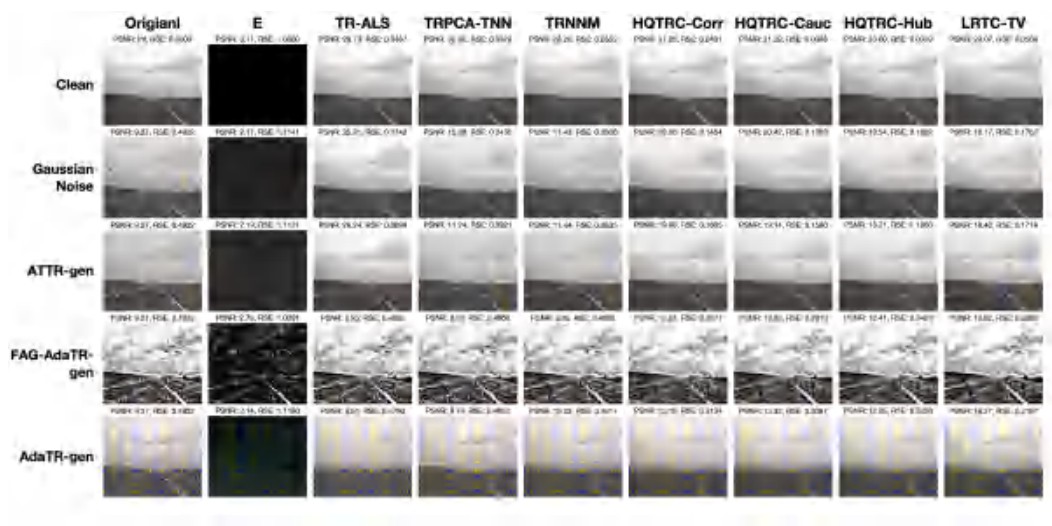

Figure 19: Visual results on tensor decomposition tasks under different attacks and defenses for the 5th frame of Video 5.

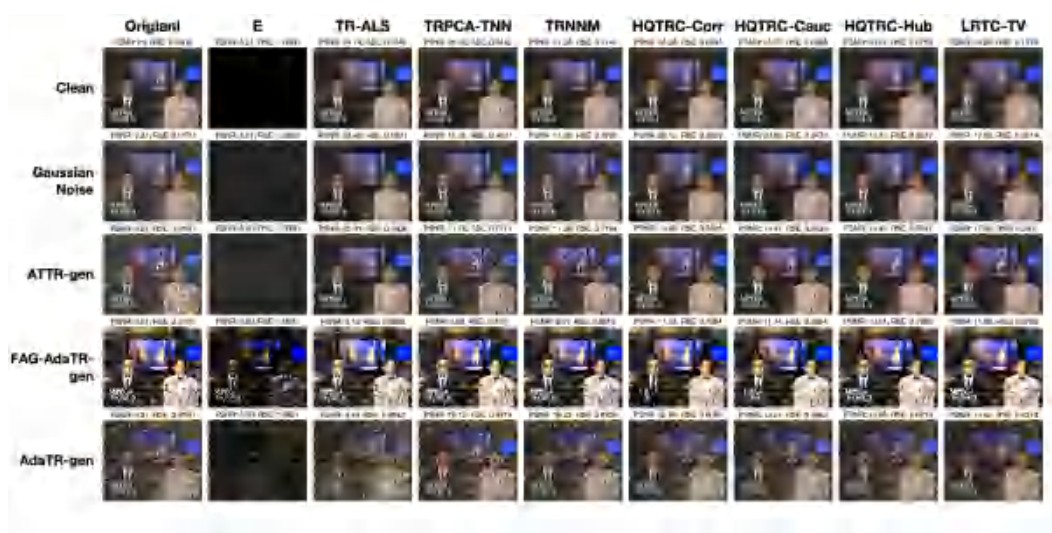

Figure 20: Visual results on tensor decomposition tasks under different attacks and defenses for the 5th frame of Video 6.

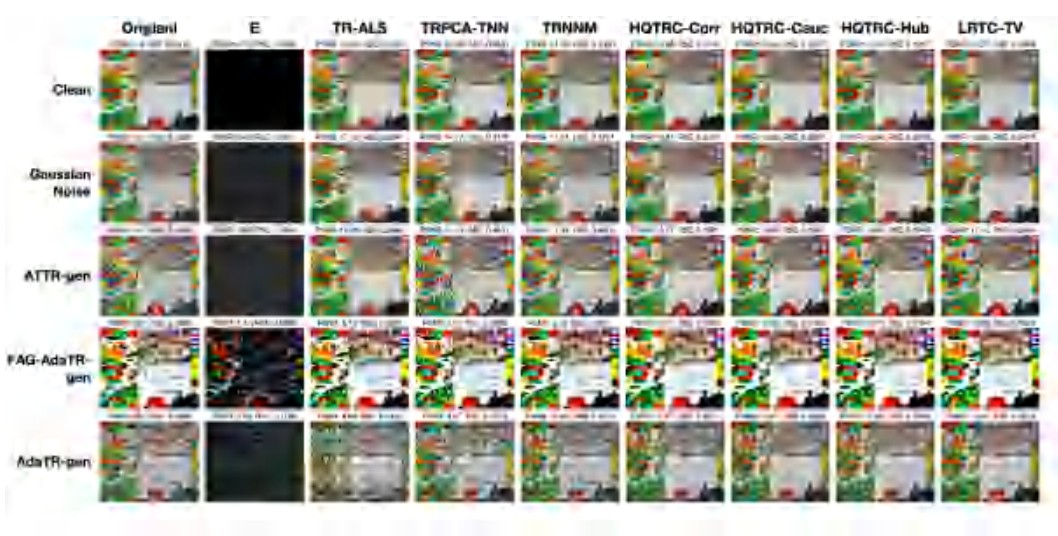

Figure 21: Visual results on tensor decomposition tasks under different attacks and defenses for the 5th frame of Video 7.

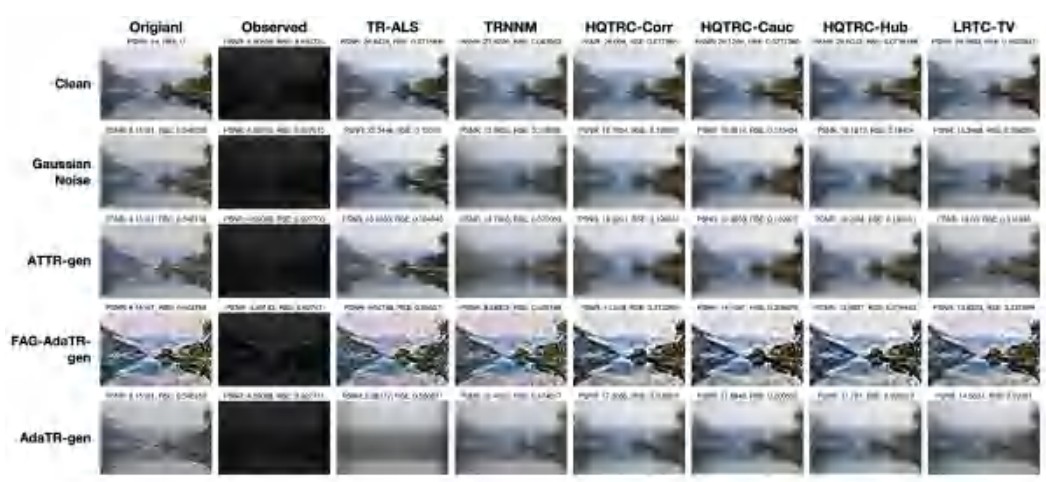

Figure 22: Additional visual results on tensor completion tasks under different attacks and defenses for Image 1 with SR=20%.

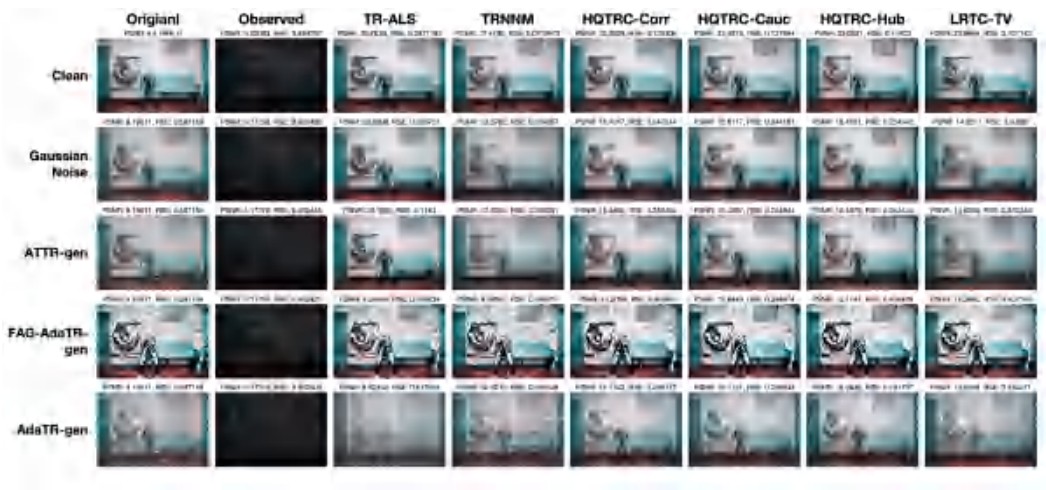

Figure 23: Additional visual results on tensor completion tasks under different attacks and defenses for Image 2 with SR=20%.

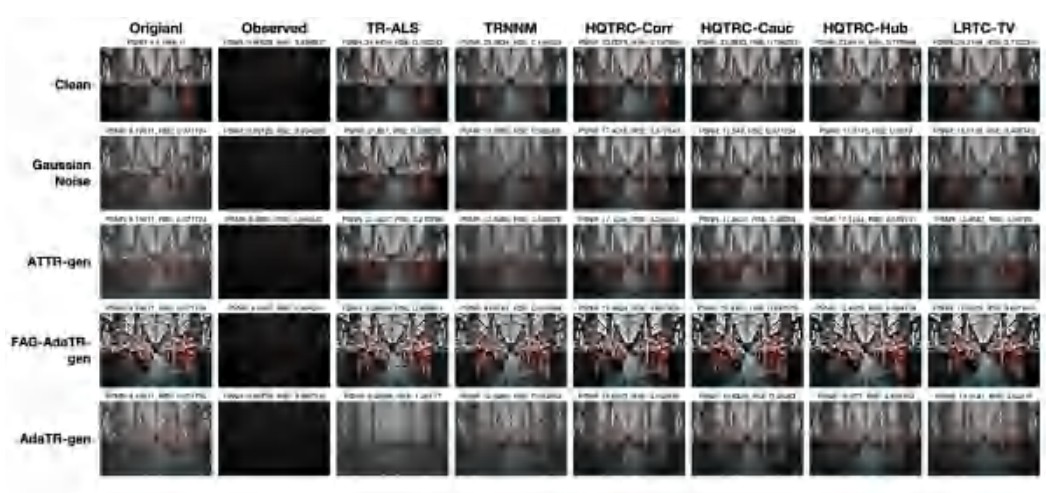

Figure 24: Additional visual results on tensor completion tasks under different attacks and defenses for Image 3 with SR=20%.

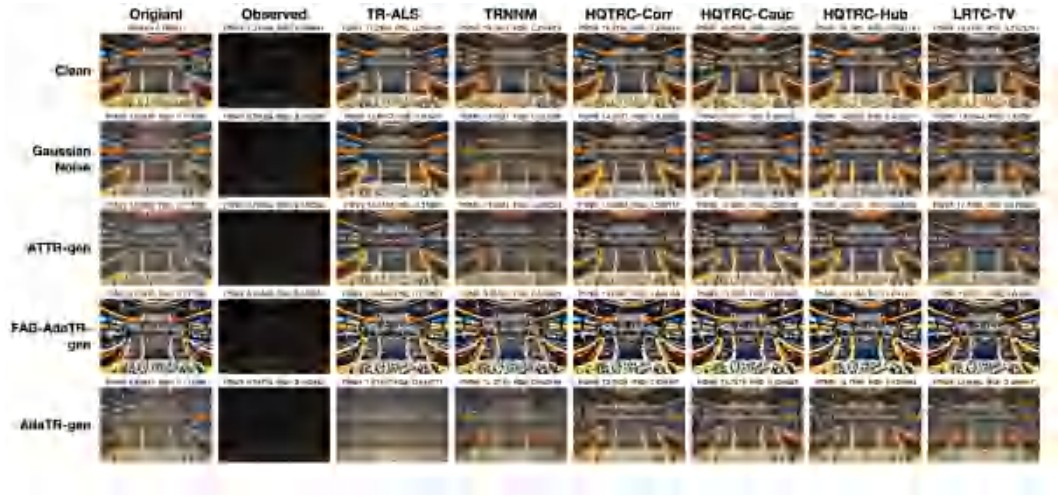

Figure 25: Additional visual results on tensor completion tasks under different attacks and defenses for Image 4 with SR=20%.

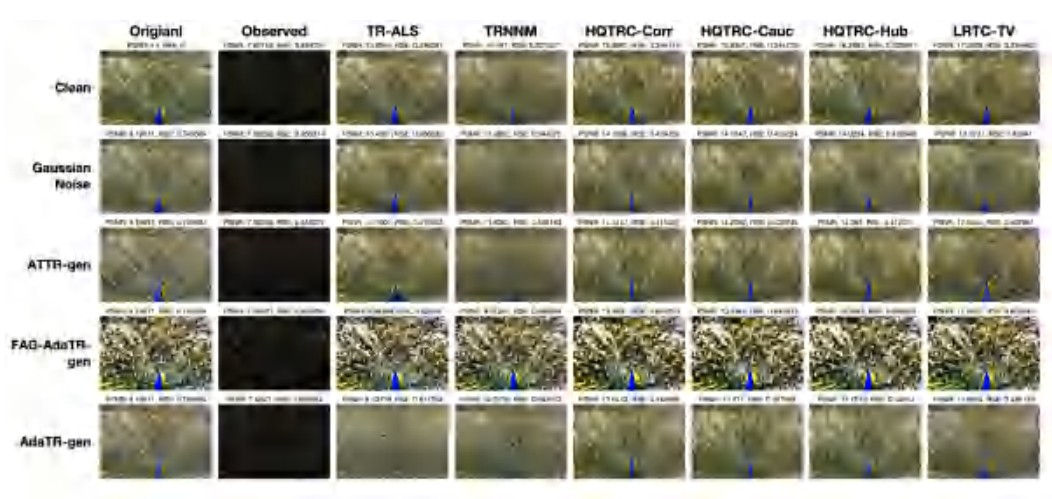

Figure 26: Additional visual results on tensor completion tasks under different attacks and defenses for Image 5 with SR=20%.

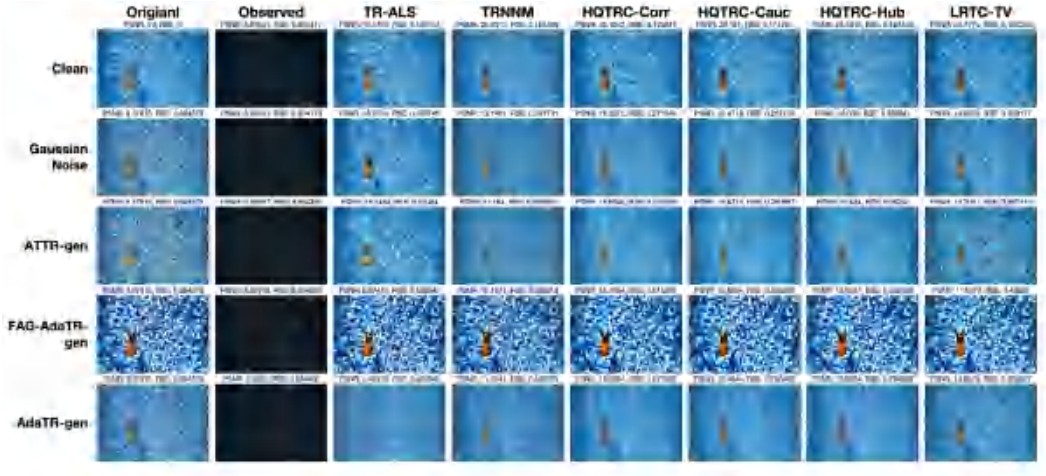

Figure 27: Additional visual results on tensor completion tasks under different attacks and defenses for Image 6 with SR=20%.

Table 11: SSIM matrix: mean $\pm$ variance across runs. Higher is better; **bold** marks the worst (lowest SSIM) per defense (column).

| Attack \ Defense | TR-ALS | TRPCA-TNN | TRNNM | HQTRC-Cor | HQTRC-Cau | HQTRC-Hub | LRTC-TV |
|---|---|---|---|---|---|---|---|
| Clean | $0.641 \pm 0.025$ | $0.896 \pm 0.000$ | $0.839 \pm 0.001$ | $0.825 \pm 0.005$ | $0.831 \pm 0.004$ | $0.863 \pm 0.001$ | $0.826 \pm 0.001$ |
| Gauss Noise | $0.572 \pm 0.010$ | $0.524 \pm 0.014$ | $0.484 \pm 0.028$ | $0.552 \pm 0.014$ | $0.557 \pm 0.012$ | $0.547 \pm 0.019$ | $0.574 \pm 0.023$ |
| ATTR-gen | $0.507 \pm 0.012$ | $0.453 \pm 0.024$ | $0.485 \pm 0.025$ | $0.596 \pm 0.025$ | $0.598 \pm 0.025$ | $0.593 \pm 0.025$ | $0.567 \pm 0.021$ |
| AdaTR-gen | $\mathbf{0.122 \pm 0.001}$ | $0.409 \pm 0.022$ | $0.409 \pm 0.023$ | $0.492 \pm 0.018$ | $0.489 \pm 0.017$ | $0.498 \pm 0.019$ | $0.515 \pm 0.021$ |
| FAG-AdaTR-gen | $0.220 \pm 0.004$ | $\mathbf{0.286 \pm 0.008}$ | $\mathbf{0.286 \pm 0.009}$ | $\mathbf{0.335 \pm 0.005}$ | $\mathbf{0.340 \pm 0.005}$ | $\mathbf{0.340 \pm 0.006}$ | $\mathbf{0.336 \pm 0.006}$ |

