# OpenReview forum: "Adversarial Attack on Tensor Ring Decomposition"
_ICLR.cc/2026/Conference — Submitted to ICLR 2026_

### Official Review · Reviewer_9zHS · 2025-10-25

**Soundness:** 3
**Presentation:** 3
**Contribution:** 3
**Rating:** 6
**Confidence:** 3

**Summary:**

This paper introduces AdaTR, an adversarial attack algorithm for Tensor Ring(TR) decomposition. It also proposes a faster variant to reduce iterative dependency called FAG-AdaTR. It first shows that directly extending the max-min formulation from ATNMF to form a baseline ATTR is not an effective attack. It shows this both empirically and theoretically. Moreover, it shows that ATR can improve performance for small perturbations. It then proposes the asymmetic max-min optimization whose objective is to directly maximize the reconstruction error of the TR decomposition. To reduce the gradient complexity and compurational cost, it proposes FAG-AdaTR. At last, it performs extensive experiments on various workloads comparing the proposesd attack with the baseline method with different defending algorithms.

**Strengths:**

The paper provides a clear theoretical analysis revealing that existing adversarial training formulations (ATTR, which is directly applied from ATNMF) can paradoxically improve tensor decomposition performance, thereby motivating the need for a stronger attack formulation.

It proposes a conceptually sound asymmetric adversarial objective that better aligns with the notion of maximizing reconstruction error and demonstrates this design through empirical results. The experiments cover diverse tasks and clearly show that the proposes attacks are substantially more destructive than exisiting baselines.

**Weaknesses:**

The paper’s main limitation is that it provides no theoretical guarantee or analysis for the proposed AdaTR and FAG-AdaTR algorithms. The only formal result concerns ATTR’s weakness, while the new methods are presented as heuristic formulations without convergence or optimality proofs. As a result, the contribution feels unbalanced.

Conceptually, AdaTR is a natural extension rather than a fundamentally new idea, and the “fast” variant is mainly an engineering improvement. The link between the theoretical critique of ATTR and the proposed algorithm is intuitive but not rigorously established.

In terms of presentation, the definition of “vulnerability” in the introduction is vague, and the distinction between adversarial training (ATTR) and attack (AdaTR) is not made clear until later. The experiments, while broad, rely only on reconstruction metrics and lack comparisons with general adversarial baselines or analysis of statistical robustness.

Typo: in line 45, it misses a citation for the ANMF paper.

**Questions:**

NA, see weakness

---

> ### Author Response · Authors · 2025-11-19
>
> # Rebuttal to Reviewer 9zHS
>
> We thank the reviewer for the detailed and constructive feedback. Below we respond to each weakness point-by-point.
>
> ---
>
> ## **Response to Weakness 1:**
> We add the new the Theorems 2 and Theorems 3 to anlysise the convergence of the proposed AdaTR and FAG-AdaTR in section 4.4. We prove that both AdaTR and FAG-AdaTR converge under mild assumptions.
>
> ## **Response to Weakness 2:**
> Thank you for the comment. We respectfully disagree with the assessment that
> AdaTR is only a natural extension of ATTR or that FAG-AdaTR is merely an
> engineering improvement.
>
> **First**, our theoretical analysis identifies a concrete structural failure of
> ATTR: it maximizes the reconstruction error of the *perturbed* tensor
> $\mathcal{X}+\mathcal{E}$, which can unintentionally *reduce* the reconstruction
> error of the clean tensor $\mathcal{X}$.
>
> **Second**, AdaTR directly addresses this issue by reformulating the attack as
> maximization of the *final clean-tensor reconstruction error*, expressed through
> the bilevel dependency $g(\mathcal{E})=\|\mathcal{X}-\hat{\mathcal{X}}(\mathcal{E})\|_F^2$.
> This is not a superficial extension of ATTR but a necessary structural change:
> AdaTR and ATTR optimize **fundamentally different objectives**, and only the
> former aligns with the intended attack goal.
>
> **Third**, the “fast” variant (FAG-AdaTR) is not merely engineering. We add the complexity analysis of the proposed FAG-AdaTR and AdaTR in Appendix G. And we also show the running time and peak memory in Appendix H.3. Regarding the FAG-AdaTR convergence analysis, we add the new Theorem 3 in the paper to support it. We even add Theorem 4 to show that its update direction remains an approximate ascent direction of the true AdaTR objective with a provably bounded deviation.
>
> We hope these clarifications make the conceptual contribution and the theoretical
> linkage more apparent.
>
>
> ## **Response to Weakness 3:**
> We have addressed each issue as follows:
>
> 1. **Clearer definition of “vulnerability.”**
>    We have now explicitly clarified the definition in Lines 90–91 of the revised manuscript:
>    *“Here, vulnerability refers to how much the reconstruction error can increase when the input tensor is perturbed under a given budget.”*
>    This makes the meaning precise and directly aligned with our problem setup.
>
> 2. **Clearer distinction between ATTR and AdaTR.**
>    To avoid confusion, we added an explicit Lemma 1 in Section 4.3 to show that the proposed AdaTR do not collapse like ATTR. Because AdaTR always increases the reconstructions error.
>    This makes the conceptual difference clear at the outset.
>
> 3. **Experiments are not limited to reconstruction metrics.**
>    While reconstruction error is a key objective for TR decomposition, our evaluation includes multiple robustness metrics across different domains:
>    - **Vision tasks:** PSNR and SSIM (Fig. 3, Sec. 5)
>    - **Recommendation systems:** RMSE, Precision@10, and Recall@10 (Tab. 2, Sec. 5)
>    These metrics go beyond reconstruction error and evaluate perceptual quality and statistical performance.
>
> 4. **Additional comparison with a general adversarial baseline.**
>    We further added a new experiment in **Appendix H.7**, comparing our approach with a *principal angle maximization* attack—a widely studied subspace-based adversarial baseline.
>    Our AdaTR and FAG-AdaTR maintain significantly stronger attack performance (Table 9), confirming the robustness and generality of our findings.
>
> We hope these clarifications fully address the reviewer’s concerns regarding presentation and experimental breadth.
>
>
> ## **Response to Weakness 4:**
> We have added the missing citation to the ANMF paper at the corresponding location in line 45 of the revised manuscript.

---

### Official Review · Reviewer_dx5X · 2025-10-30

**Soundness:** 2
**Presentation:** 3
**Contribution:** 2
**Rating:** 4
**Confidence:** 5

**Summary:**

This paper studies the adversarial attacks on tensor ring decomposition. It is first observed that the classical symmetric min-max method (ATTR) may even improve the performance of the model under certain conditions. Therefore, it motivates to development of the asymmetric method based on the bilevel optimization. Since the proposed method requires complicated gradient computation with backpropagation, a simplified version of the algorithm is proposed by deactivating some variables’ gradients (FAG-AdaTR). Experiments on color image decomposition attacks, video decomposition attacks, tensor completion, and recommendation systems are presented to show the effectiveness of the proposed method.

**Strengths:**

(1) The writing is clear, and readers can easily follow.

(2) The experiments are comprehensive. It covers various tensor data, including color images, videos, recommendation systems, and general tensor data.

**Weaknesses:**

(1) I do not quite agree that the max-min problem (5) is symmetric and the bilevel form (10) is asymmetric. In minimax optimization, if we do not assume the Nash equilibrium (or some other conditions such as a strongly convex and strongly concave objective), the order of min and max cannot be changed. Therefore, (5) is asymmetric. I mean (5) is exactly equivalent to (10) if the order of min and max cannot be changed. In (5), given E, G is selected to minimize the objective. Therefore, the main motivation and claims in this paper are not correct from a minimax optimization perspective.

(2) In your algorithm, you deactivate many variables’ gradients (w.r.t. E) for simplifying the computation. However, does it still guarantee the convergence of the algorithm? Is the simplified gradient still a descent direction? Is it possible that after you mask some gradients, the simplified gradient is not valid for the problem? I am suspecting the effectiveness of the simplified algorithm, at least theoretically.

(3) The paper lacks a theoretical analysis of the algorithm (FAG-AdaTR). Is your algorithm convergent? If yes, what kind of point does it converge to?

**Questions:**

see weaknesses

---

> ### Author Response · Authors · 2025-11-19
>
> # Rebuttal to Reviewer dx5X
>
> We thank the reviewer for the detailed and constructive feedback. Below we respond to each weakness point-by-point.
>
> ---
>
> ## **Response to Weakness 1:**
>
> Thank you for the clarification. We agree that in general minimax problems do
> not allow exchanging the order of $\max$ and $\min$ unless additional convexity
> or Nash–type conditions hold. Our intention was not to claim that problem (5)
> is “symmetric’’ in the minimax-theoretic sense, nor that (5) and (10) differ by
> a change in the order of optimization.
>
> Our motivation is instead based on the observation that **ATTR and AdaTR
> optimize fundamentally different objectives**, even though both can be written
> in a max–min form. Specifically:
>
> - In (5), ATTR maximizes the reconstruction error of the *perturbed*
>   tensor $\mathcal{X} + \mathcal{E}$; the inner minimization selects
>   $\mathcal{G}$ that best reconstructs $\mathcal{X} + \mathcal{E}$.
>
> - In (10), AdaTR maximizes the reconstruction error of the *clean* tensor
>   $\mathcal{X}$ **after** TR decomposition under perturbation, i.e.,
>   $g(\mathcal{E}) = \|\mathcal{X} - \hat{\mathcal{X}}(\mathcal{E})\|_F^2$.
>   The bilevel formulation explicitly isolates this dependency through
>   $\mathcal{G}^{*}(\mathcal{E})$.
>
> Therefore, although (5) and (10) share the same max–min order, they are **not
> equivalent**: the outer maximizes solving two different problems
> (worst TR reconstruction of $\mathcal{X}+\mathcal{E}$ vs. reconstruction of
> $\mathcal{X}$ using factors obtained from $\mathcal{X}+\mathcal{E}$). This
> The difference is exactly what causes AdaTR not to collapse like ATTR (Lemma 1 and Theorem 2).
>
> We have revised the motivation at the end of the Introduction section.
>
>
>
> ## **Response to Weakness 2:**
> In the revised manuscript, we explicitly address these questions through new
> theoretical results:
>
> 1. **Convergence of AdaTR and FAG-AdaTR (Theorems 2 and 3).**
>    We prove that both AdaTR (the full-gradient method) and FAG-AdaTR
>    (the simplified-gradient method) converge under mild assumptions.
>    The analysis shows that masking the dependence of intermediate TR-ALS
>    iterates on $\mathcal{E}$ does not break the convergence mechanism.
>
> 2. **Validity of the simplified gradient (Theorem 4).**
>    We further provide a quantitative bound on the deviation between the
>    simplified gradient used by FAG-AdaTR and the true AdaTR gradient.
>    Specifically, for the solution $\mathcal{E}^\star$ produced by FAG-AdaTR,
>    the inner product with the true gradient satisfies
>    $$
> \big\langle \nabla g(\mathcal{E}^\star),\ \mathcal{E}^\star - \mathcal{E}^{(t)} \big\rangle \ge - \sqrt{\epsilon}\ \varepsilon_g,
>    $$
>
>    This shows that the simplified update direction cannot become strongly
>    negative, i.e., **the direction used by FAG-AdaTR is still a valid
>    ascent direction for the AdaTR objective**, and cannot invalidate the
>    optimization dynamics.
>
> Together, these new theoretical results guarantee that the simplified gradient
> does not compromise convergence and remains a valid ascent direction.
>
>
> ## **Response to Weakness 3:**
> We have already added a theoretical analysis of
> FAG-AdaTR in the revised manuscript. As shown in **Theorem 3**, the algorithm
> is convergent, and **Theorem 4** further characterizes the limit point as an
> approximate stationary point of the true AdaTR objective with a quantified
> gradient deviation bound.

---

### Official Review · Reviewer_HDzR · 2025-11-01

**Soundness:** 3
**Presentation:** 3
**Contribution:** 3
**Rating:** 4
**Confidence:** 4

**Summary:**

The paper studies adversarial attacks on TR decomposition. The authors argue that the conventional minmax ATTR objective (maximize w.r.t. perturbation, then minimize w.r.t. TR factors) can unintentionally improve low-rank reconstruction under small budgets and therefore isn't a true "attack" on TR (Thm 1). To address this, the authors propose an asymmetric bilevel formulation (AdaTR): minimize TR factors on the perturbed tensor but maximize the reconstruction error on the original tensor in Eq. (10). The perturbation is obtained through the ALS updates. The paper also introduces a faster version in which a closed-form approximate gradient for each mode is derived. Experiments on images, videos, and a recommender show larger degradation vs. ATTR across several TR-based defenses.

**Strengths:**

+ The paper identifies a failure mode of ATTR for small budgets.
+ The bilevel objective is an interesting formulation aligning with the attack goal.
+ Fast approximate variant (FAG-AdaTR) with explicit gradients.
+ Good experimental evaluation (images, videos, completion, recommender).

**Weaknesses:**

- The problem lacks clear motivation and lacks clarity on the threat model. The attack norm is Frobenius on the full tensor (global energy budget). For vision tasks this may not be aligned with perceptual threat models. For recommendation it's nontrivial what perturbing all entries means. The paper would benefit from a precise threat model per application domain (who controls what, where noise is injected, etc).
- While Theorem 1 shows ATTR's potential to help at small $\varepsilon$, the intuition behind how the proposed formulation in Eq. (10) fixes this is lacking. It is more procedural than structural. A theorem or a lemma explaining why the AdaTR objective directly targets the final error (and can't collapse as with ATTR) would strengthen the story. You can contrast the two objectives' gradients w.r.t E to make the fix more convincing. Right now the argument is mostly empirical (Fig. 1).
- If I understood correctly, the FAG-AdaTR efficiency comes from decoupling E from some iterates trading bias for speed. However, the approximation error (how far from the true gradient ascent) is not quantified, and there is no theoretical attack optimality.
- Beyond the approximation shortcut, there is no complexity analysis unless I missed that.
- The paper mainly compares to ATTR and Gaussian noise, plus defense methods designed for completion/denoising, not attack methods on TR. A comparison to projected gradient attacks on the low-rank objective or to adversarial subspace attacks adapted from matrices would be important and show the gains.
- For recommendation, it is unclear whether attacks respect the typical sparsity. Perturbing the dense rating tensor can be unrealistic. The setup should align with feasible manipulations (such as limited user/item edits).

The paper makes a worthwhile point (ATTR can help rather than harm for small $\varepsilon$) and proposes a nice bilevel objective with a fast approximation. However, there are many weaknesses that need to be addressed to make this a solid paper. For ICLR, I'd want to see a crisper formal contrast between ATTR and AdaTR, compute/scaling and approximation-error analysis for FAG, models aligned per application. If strengthened along these lines, the paper could be compelling.

**Questions:**

1) What is the adversary's capability per domain (images/videos vs recommender, etc)? Why Frobenius norm on the entire tensor, and how would results change with other realistic constraints?
2) Can you provide a theoretical statement showing that the AdaTR gradient aligns with maximizing the final reconstruction error, whereas ATTR can degenerate to maximizing $\Delta E$ as claimed?
3) Can you bound or study the bias introduced by the FAG? When does it deviate most from AdaTR?
4) What are the peak memory and time vs. tensor size, rank, and inner ALS iterations for AdaTR and what speedup does FAG deliver in practice?
5) Comparisons to other attack methods?

---

> ### Author Response · Authors · 2025-11-19
>
> # Rebuttal to Reviewer HDzR
>
> We thank the reviewer for the detailed and constructive feedback. Below we respond to each weakness point-by-point.
>
> ---
>
> ## **Response to Weakness 1:**
> We have clarified the threat model and the motivation more explicitly in the revised manuscript:
> 1. Clearer motivation added to the main text (p.2):
>  This clarifies the practical impact of adversarial perturbations on different downstream applications.
>
> 2. New experiments comparing Frobenius vs. $L_\infty$ constraints (Appendix G.6).
>
> 3. Clarification of the recommendation-system threat model:
>  We have updated the section 5.4 to explicitly state that the perturbation is restricted to the available rating entries, making the threat model realistic for this domain.
>
> ---
>
> ## **Response to Weakness 2:**
>  In the revised version, we have added a new theoretical result (Lemma 1, following Theorem 2) in section 4.4. This Lemma provides the theoretical justification requested by the reviewer: the proposed AdaTR directly optimizes the intended objective and cannot collapse as ATTR does.
>
> ---
>
> ## **Response to Weakness 3:**
> In the revised manuscript we have added a new result (Theorem 4) that directly addresses this concern.
>
> While FAG-AdaTR indeed improves efficiency by decoupling $\mathcal{E}$ from intermediate TR-ALS iterates, Theorem 4 provides a formal quantification of the resulting approximation error.
>
> ---
>
> ## **Response to Weakness 4:**
> Thank you for pointing this out. We have now added a dedicated complexity analysis for both **AdaTR** and **FAG-AdaTR** in **Appendix G**. We hope this addition addresses the reviewer’s concern regarding the lack of complexity analysis.
>
> ---
>
> ## **Response to Weakness 5:**
> We add the new experiments in the Appendix H.7 to evaluate our methods under a recently–considered adversarial strategy based on **principal angle maximization**.
>
>
> ---
>
> ## **Response to Weakness 6:**
> We apologize for not making this sufficiently clear in the original submission.
> In our recommendation experiments, **all perturbations are applied only to the observed entries**. We have updated the section 5.4 to explicitly state that the attack is restricted to the observed ratings.
>
> ---
>
> # **Responses to Reviewer’s Specific Questions**
> Below we address each explicit question raised by the reviewer.
>
> ---
>
> ### **Q1. Response:**
>
> **Adversarial capability per domain.**
> Our threat model aligns with what is feasible in each domain.
> For image/video tensors, perturbations typically correspond to pixel-level changes, where an $\ell_\infty$ constraint is standard.
> For recommendation data, perturbations correspond to editing a subset of observed user–item interactions, where Frobenius-norm–based budgets naturally capture global rating changes while preserving sparsity.
>
> **Why Frobenius norm?**
> The Frobenius norm provides a global energy budget that yields stronger degradation of factorization quality. However, we agree that other constraints may be more realistic for visual data.
>
> **Additional constraints.**
> To address this, we have added an $\ell_\infty$-constrained attack experiment in **Appendix H.5**.
>
> ---
>
> ### **Q2. Response:**
>
> Yes. We now provide a Lemma 1 statement confirming this behavior in the section 4.4.
>
> ---
>
> ### **Q3. Response:**
>
> We now address them theoretically in the revised manuscript.
>
> **(1) Bounding the bias introduced by FAG.**
> In **Theorem 4**, we provide an explicit upper bound on the deviation of the FAG-AdaTR solution from the true AdaTR gradient. Formally, for the perturbation $\mathcal{E}^\star$ returned by FAG-AdaTR, we show that
>
> $$
> \big\langle \nabla g(\mathcal{E}^\star),\ \mathcal{E}^\star - \mathcal{E}^{(t)} \big\rangle \ge - \sqrt{\epsilon}\ \varepsilon_g,
> $$
>
> This inequality ensures that the gradient of the true AdaTR objective **cannot be strongly negative** at the FAG solution, i.e., the bias introduced by FAG is *explicitly bounded* and the update direction remains approximately ascent-valid.
>
> **(2) When FAG deviates most from AdaTR.**
> The same bound implies that deviation is largest when the decoupling assumption is least accurate—namely, when the TR factors change rapidly across inner TR-ALS sweeps.
> In such cases, ignoring the dependence of intermediate TR factors on $\mathcal{E}$ increases $\varepsilon_g$, leading to a looser ascent condition.
>
>
>
> ---
>
> ### **Q4. Response:**
>
> We have now explicitly measured both **peak memory** and **runtime** for AdaTR and FAG-AdaTR as functions of the same tensor size, TR-rank, and the number of inner ALS iterations. The results are reported in **Appendix H.3**.  These results show that the FAG-AdaTR is almost 2 times faster than AdaTR and 10 times lower peak memory than AdaTR.
>
> ---
>
> ### **Q5. Response:**
> We add the new experiments in the Appendix H.7 to evaluate our methods under a recently–considered adversarial strategy based on **principal angle maximization** in matrix subspace.
>
> ---

---

### Official Review · Reviewer_CGTS · 2025-11-02

**Soundness:** 3
**Presentation:** 2
**Contribution:** 2
**Rating:** 4
**Confidence:** 3

**Summary:**

The paper investigates a previously unexplored question: are tensor decompositions vulnerable to adversarial perturbations? Focusing on Tensor-Ring (TR) decomposition, the authors formulate a dedicated adversarial attack on TR and derive a convergent gradient-based solver for the attacker.

**Strengths:**

1. This work proposes the first adversarial attack tailored to tensor decomposition; prior work (ATNMF, LaFa) targets matrix factorisation or poisons data, not the decomposition operator itself.
2. Additionally, it proposes a novel asymmetric bilevel formulation that flips the usual min-max adversarial-training order, directly optimising the attacker’s goal (max reconstruction error).
3. The experiments are conducted in various applications and show the universality of the method.

**Weaknesses:**

1. This paper claims “tensor decomposition” vulnerability, but only TR-ALS is attacked; it is unclear whether fragility extends to CP, Tucker, TT, or SVD-based methods. The authors may run the same asymmetric objective on Tucker-ALS and TT-SVD (only requires swapping the composition operator).
2. This paper should compare more baselines. ATTR is a natural extension of ATNMF; however, data-poisoning attacks or subspace-rotation attacks are also relevant but omitted. The authors should include a subspace-rotation attack baseline that maximises principal-angle deviation; this checks whether TR fragility is simply due to low-rank bias rather than the proposed bilevel formulation.
3. The experiments don't contain simple defenses. This work does not investigate whether adversarial training or input denoising can mitigate perturbations, leaving practitioners without effective countermeasures. The authors should add a defensive experiment: wrap TR-ALS with adversarial training using the proposed attack.

**Questions:**

Please refer to the weaknesses.

**Details Of Ethics Concerns:**

None.

---

> ### Author Response · Authors · 2025-11-19
>
> # Rebuttal to Reviewer CGTS
>
> We thank the reviewer for the constructive feedback and for recognizing the novelty and contributions of our work. Below we respond to each weakness and clarify the key concerns.
>
> ---
>
> ## **Response to Weakness 1:**
>
>
> **Response:**
> We thank the reviewer for pointing out this important question regarding the generality of our findings beyond TR-ALS. Following the suggestion, we have extended our experiments to Tucker-ALS, CP-ALS, and TT-ALS by directly replacing the TR decomposition operator in our asymmetric bilevel objective with the corresponding multilinear operators. The attack formulation remains unchanged, and no algorithmic redesign is required.
>
> New results added to Appendix G. 4:
> 1. Tucker-ALS, CP-ALS, and TT-ALS exhibit the same vulnerability pattern as TR-ALS.
> 2. This confirms that the fragility is not specific to TR-ALS, but is inherent to low-rank tensor decompositions driven by least-squares reconstruction.
>
> Regarding TT-SVD: TT-SVD is not an ALS method but a direct one-pass SVD-based algorithm and therefore does not fit the asymmetric bilevel formulation we analyze. Our claims and experiments focus on ALS-driven decompositions, which are clarified in the revised manuscript. But in color image, video experiments, we have used SVD-based TR method to defend the AdaTR attack, such as TRPCA, TRNNM, and HQTRC. The results show that the proposed methods still have the best attack performance.
>
> We thank the reviewer again for the helpful comment, which led us to extend the empirical scope and clarify the generality of our claims.
>
> ---
>
> ## **Response to Weakness 2:**
>
> **Response:**
> We appreciate the reviewer’s suggestion regarding baselines. To the best of our knowledge, there are currently no established adversarial-perturbation baselines for tensor decomposition. Our method is, to our knowledge, the first attack targeting tensor decomposition models. Even ATTR, which the reviewer mentions, is not an existing method—it is a variant we derived by extending ATNMF to the tensor-ring setting. Thus, we have already included every baseline that exists or can be meaningfully adapted.
>
> Regarding “subspace-rotation attacks”:
> We add the new experiments in the Appendix H.7 to evaluate our methods under a recently–considered adversarial strategy based on **principal angle maximization**. This attack is inspired by classical subspace-based adversarial analysis, where the adversary seeks a rank-one perturbation $\Delta \mathbf{X} = \mathbf{a}\mathbf{b}^\top$ that maximizes the largest principal angle between the clean feature subspace and the perturbed feature subspace.
>
> Although the principal-angle maximization attack is effective in shifting the underlying subspace, our methods still significantly outperform it. The reason is that the subspace deviation induced by principal-angle maximization does **not** necessarily correspond to a large reconstruction error. Principal angles measure the worst-case discrepancy between subspaces, but they do not directly control how the corrupted features affect pixel-level reconstruction. In contrast, both AdaTR and FAG-AdaTR are designed to minimize the **actual reconstruction error**, and thus remain robust even when the adversary succeeds in enlarging the principal angle.
>
> ---
>
> ## **Response to Weakness 3:**
>
>
> **Response:**
> We thank the reviewer for raising the question of defenses. We apologize for the lack of clarity in the main text—our experiments do evaluate several defense-like mechanisms, but this was not made explicit enough. Specifically, in the image and video reconstruction/completion tasks, all compared algorithms (e.g., cartoon-denoising priors, total-variation–based completion, sparse regularizers) already incorporate techniques that are designed precisely to suppress noise. In Appendix G.1, we provide a detailed description of these methods and explain which algorithms serve as attack algorithms and which serve as defense algorithms.
>
> Unlike neural networks, tensor decomposition methods such as TR/CP/Tucker are not parametric models that retain memory of past samples. ALS-based algorithms recompute factor matrices from scratch for every new input, thus making conventional adversarial training (i.e., learning from adversarial examples across epochs) inapplicable. We attempted a form of “adversarial initialization training,” where the initial core tensors $[\mathcal{G}^{(0)}]$ were optimized to minimize the adversarial objective, but this approach did not yield meaningful robustness—confirming that standard adversarial training does not transfer to nonparametric tensor decompositions.

---

### Author Response · Authors · 2025-12-03

# Rebuttal Summary

We thank the reviewers for their careful evaluations and constructive feedback. Below we summarize how the revised manuscript addresses the main concerns and strengthens the contribution.

---

## 1. Why AdaTR Can not Collapse like ATTR

Reviewers HDzR, dx5X, and 9zHS asked for a more theoretical contrast between ATTR and AdaTR.

We added new Theorem 2 and Lemma 1 in the manuscript to support this. Theorem 2 shows that the proposed AdaTR is convergent under mild assumptions. Based on Theorem 2, we further give Lemma 1 to show that AdaTR can not collapse like ATTR. The details can be found in section 4.4 of the revised manuscript.




---

## 2. Validity and Convergence of FAG-AdaTR

Reviewers HDzR, dx5X, and 9zHS were concerned that FAG-AdaTR might break convergence or lead to invalid gradients.

We addressed this by adding:

- **Theorem 3:** Proves convergence of FAG-AdaTR under mild smoothness and step-size assumptions.
- **Theorem 4:** We provide a quantitative bound on the deviation between the simplified gradient used by FAG-AdaTR and the true AdaTR gradient. This shows that the direction used by FAG-AdaTR is still a valid ascent direction for the AdaTR objective.

The details can be found in section 4.4 of the revised manuscript.

---

## 3. Generality Beyond Tensor Ring (TR) Decomposition

Reviewer CGTS asked whether the observed vulnerability is specific to TR-ALS. We extended our experiments by directly replacing the TR operator in our bilevel formulation with Tucker-ALS, CP-ALS, and TT-ALS. All these ALS-based tensor decompositions exhibit the same vulnerability pattern under our attack. The details can be found in Appendix H.4 of the revised manuscript.

---

## 4. Additional Attack Baseline Experiments (Principal-Angle Maximization Attacks)

Reviewer CGTS and HDzR suggested comparing against subspace-rotation or principal-angle–based attacks.

We implemented a principal-angle maximization attack method, where the adversary maximizes the largest principal angle between the clean and perturbed tensor subspaces.
This attack does distort the subspace but does not lead to large reconstruction errors. In contrast, AdaTR and FAG-AdaTR consistently induce much larger reconstruction degradation across all tasks.

Specifically, the principal-angle maximization is the attack method in the subspace. Our asymmetric objective is the attack method in the low-rank reconstruction of the original data. The details can be found in Appendix H.8 of the revised manuscript.

---

## 5. Complexity and Practical Efficiency

Reviewer HDzR found that the complexity analysis is missing, and is also concerned about the peak memory and time of the proposed methods.

We added a detailed complexity analysis in Appendix G of the revised manuscript. We also reported the runtime and peak memory comparisons for images and videos in Appendix H.3; these results show that FAG-AdaTR reduces peak memory usage by almost 10 times that of AdaTR and is 2 times faster than AdaTR, while maintaining strong attack performance.

---

## 6. Overall Assessment

The revised submission now includes:

- 2 New theorems (Theorem 2, 3) for **AdaTR** and **FAG-AdaTR** are convergent, 1 new lemma (Lemma 1) supports that AdaTR can not collapse like ATTR, 1 new theorem (Theorem 4) shows the FAG-AdaTR is still a valid algorithm,
- New adversarial attack experiments on **Tucker/CP/TT** decompositions in Appendix H.4,
- A new **principal-angle maximization** attack baseline,
- A clarified, realistic **threat model** for recommender systems domain in section 5.4,
- Detailed **complexity** analysis in Appendix G, **runtime and peak memory** results are given in Appendix H.3,
- Additional experiments under different norms in Appendix H.6.

We believe these additions directly address all reviewer concerns and substantially strengthen both the **theoretical** and **empirical** contributions of the paper.

---

### Meta-Review · Area_Chair_4NoU · 2025-12-30

**Summary:**

An adversarial attack algorithm for Tensor Ring (TR) decomposition is proposed. The proposed approach is a heuristic without much theoretical support (at least in the original submission).

**Reviewer Concerns:**

A significant amount of theoretical claims are added after the rebuttal. No reviewer assessed their correctness (and understandably, this is not their responsibility; the theoretical claims should have been included in the first submission).

**Reviewer Scores:**

There is a slight chance that some scores would have been higher if the theoretical claims were included in the first submission. However, too much of them are added during the rebuttal period, which is not good practice for the reviewers to fully assess them.

---

### Decision · Program_Chairs · 2026-01-26

Reject